# Distinct protocerebral neuropils associated with attractive and aversive female-produced odorants in the male moth brain

Jonas Hansen Kymre[1†], XiaoLan Liu[2,3†], Elena Ian[1], Christoffer Nerland Berge[1], GuiRong Wang[3], Bente Gunnveig Berg[1], XinCheng Zhao[2]*, Xi Chu[1]*

[1]Chemosensory lab, Department of Psychology, Norwegian University of Science and Technology, Trondheim, Norway; [2]Department of Entomology, College of Plant Protection, Henan Agricultural University, Zhengzhou, China; [3]State Key Laboratory for Biology of Plant Disease and Insect Pests, Institute of Plant Protection, Chinese Academy of Agricultural Sciences, Beijing, China

**Abstract** The pheromone system of heliothine moths is an optimal model for studying principles underlying higher-order olfactory processing. In *Helicoverpa armigera*, three male-specific glomeruli receive input about three female-produced signals, the primary pheromone component, serving as an attractant, and two minor constituents, serving a dual function, that is, attraction versus inhibition of attraction. From the antennal-lobe glomeruli, the information is conveyed to higher olfactory centers, including the lateral protocerebrum, via three main paths – of which the medial tract is the most prominent. In this study, we traced physiologically identified medial-tract projection neurons from each of the three male-specific glomeruli with the aim of mapping their terminal branches in the lateral protocerebrum. Our data suggest that the neurons' widespread projections are organized according to behavioral significance, including a spatial separation of signals representing attraction versus inhibition – however, with a unique capacity of switching behavioral consequence based on the amount of the minor components.

*For correspondence:
xincheng@henau.edu.cn (XCZ);
xi.chu@ntnu.no (XC)

[†]These authors contributed equally to this work

## Introduction

Olfactory circuits serve a central role in encoding and modulating sensory input from the natural surroundings. Understanding how these chemosensory circuits translate signals with different hedonic valences into behavior is an essential issue in neuroscience. With a relatively simple brain and a restricted number of associated odors evoking opposite innate behaviors, that is, attraction and aversion, the insect pheromone pathway is an optimal system to address this question. In moth, pheromone-evoked behaviors are linked to a hardwired circuit in the lateral protocerebrum, including the lateral horn (*Insect Brain Name Working Group et al., 2014*; *Martin et al., 2011*). This brain region shares many neural principles with the mammalian cortical amygdala (*Miyamichi et al., 2011*; *Sosulski et al., 2011*). In contrast to the semi-random neuronal connectivity in another higher-order olfactory center of the insect, the mushroom body calyx (mammalian piriform cortex analog; *Su et al., 2009*), the neuronal wiring in the lateral protocerebrum is characterized by a form of spatial clustering relying on behavioral significance (for details, see *Figure 1A*). Signals inducing different behaviors, such as pheromones versus food odors, are segregated in the lateral protocerebrum both in fruit fly and moths (*Grabe and Sachse, 2018*; *Homberg et al., 1988*; *Zhao et al., 2014*). In the fruit fly, such spatial separation is shown to include even attractive versus repulsive odor signals (*Grabe and Sachse, 2018*; *Min et al., 2013*). Thus, at the level of the lateral protocerebrum, it

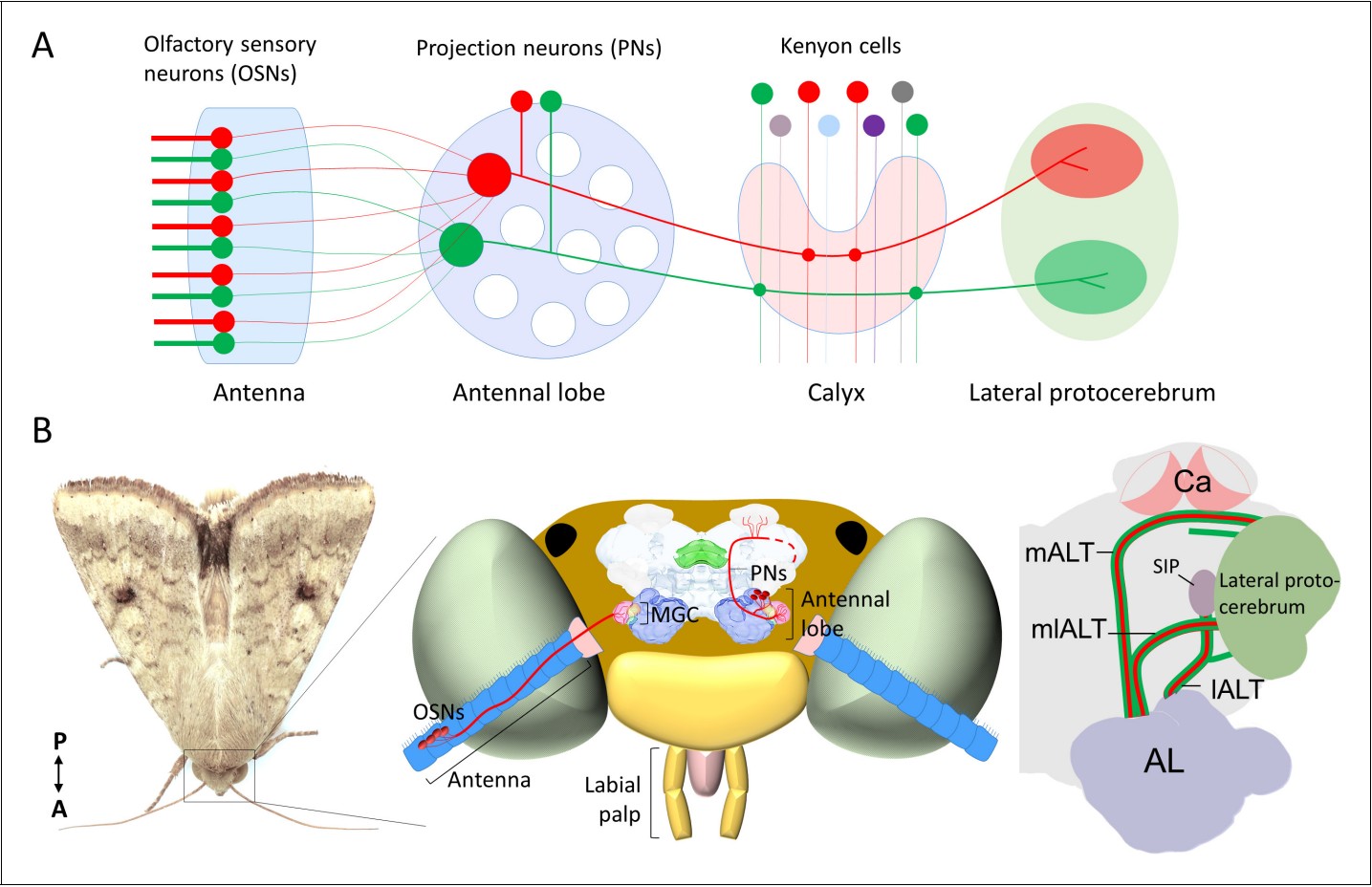

**Figure 1.** Schematic representation of the neurons in the insect olfactory system. (**A**) All olfactory sensory neurons (OSNs) that express the same odorant receptor project their axons to the same antennal-lobe (AL) glomerulus. AL projection neurons (PNs) passing along the most prominent tract, the medial AL tract (mALT) have dendrites in a single glomerulus and receive direct input from one OSN class. Neurons tuned to food odors versus pheromones are portrayed in *green* and *red*, respectively. The medial-tract PNs first project to the calyces of the mushroom body (Ca), where they contact Kenyon cells. The neuron connectivity in Ca has a semi-random topographical distribution, where signals are segregated according to the nature of sensory input, for example, pheromone and general odor PNs have their terminals located at the inner and outer part of calyces, respectively (*Zhao et al., 2014*), but connections between the projection neurons and the Kenyon cells are random (*Caron et al., 2013*). Instead of encoding the identity of stimuli, the Kenyon cells process signals of positive versus negative valence into different terminal clusters (*Aso et al., 2014a*; *Aso et al., 2014b*). After innervating the Ca, the medial-tract PNs project into distinct sub-regions in the lateral protocerebrum according to the behavioral significance of the relevant odor signal. The PNs in other antennal-lobe tracts (ALTs) are not shown in this panel. (**B**) Schematic presentation of the olfactory system of *H. armigera*. Left: Portrait of a *H. armigera* male. Middle: Outline of the head capsule in dorsal view with the main olfactory organ (antenna) and AL along with OSNs and PNs tuned to the primary sex pheromone. The medial-tract PNs indicated here are uniglomerular neurons connecting the AL with the Ca and the lateral protocerebrum. The distinct protocerebral regions innervated by these medial-tract pheromone-sensitive PNs are undefined. Right: Schematic drawing of the main parallel ALTs: medial, mediolateral, and lateral ALT (mALT, mlALT, and lALT). Here, we followed the naming system used in *Insect Brain Name Working Group et al., 2014*. The mALT was previously named the inner antenno-prorocerebral tract. The mlALT, previously termed the middle antenno-protocerebral tract, is formed by multiglomerular PNs targeting the lateral protocerebrum directly. The lALT, previously named as outer antenno-protocerebral tract, is formed by both uni- and multi-glomerular PNs. In moths, some of the PNs confined to the lALT target the superior intermediate protocerebrum (SIP) and only a small part of them project into the Ca via lateral protocerebrum like the lALT PNs in honeybee/ant (*Galizia and Rössler, 2010*; *Homberg et al., 1988*; *Ian et al., 2016*).

appears that odor cues are represented in different widespread sub-domains that display a form of spatial pattern according to behavioral significance, including valence.

In the moth antennal lobe (AL; mammalian olfactory bulb analog), the male-specific macroglomerular complex (MGC) receives input from olfactory sensory neurons (OSNs, *Figure 1B*). The MGC glomeruli process input about a few female-produced signals, of which one primary constituent acts as an unambiguous attractant, while others often enhance attraction at low doses but serve as behavioral antagonists at higher doses (*Chang et al., 2017*; *Gothilf et al., 1978*; *Kehat and Dunkelblum,*

*1990*; *Wu et al., 2015*; *Zhang et al., 2012*; *Martin et al., 2013*). In addition to composition of the pheromone blend, other aspects such as plume structure, duration, and concentration are also very relevant for the activity of MGC output neurons (*Lei and Hansson, 1999*). From the MGC, the pheromone signals are conveyed to higher brain centers, including the lateral protocerebrum, by male-specific projection neurons (PNs) following three main tracts (*Figure 1B*), that is, the medial, medio-lateral, and lateral ALT (mALT, mlALT and lALT, respectively [*Homberg et al., 1988*; *Lee et al., 2019*]). The most prominent is the mALT, resembling the mammalian olfactory tract by consisting of uniglomerular PN axons (reviewed by *Lledo et al., 2005*). Although a considerable number of medial-tract MGC neurons have previously been reported in various moth species (*Anton et al., 1997*; *Berg et al., 1998*; *Christensen and Hildebrand, 1987*; *Christensen et al., 1991*; *Christensen et al., 1995*; *Hansson et al., 1994*; *Hansson et al., 1991*; *Jarriault et al., 2009*; *Kanzaki et al., 1989*; *Kanzaki et al., 2003*; *Nirazawa et al., 2017*; *Seki et al., 2005*; *Vickers et al., 1998*; *Zhao and Berg, 2010*; *Zhao et al., 2014*), the main focus has so far been odor coding within the MGC. Only a few studies have paid close attention to the protocerebral projections of these medial-tract PNs (*Kanzaki et al., 2003*; *Seki et al., 2005*; *Zhao et al., 2014*). Thereby, our two main questions are where in the lateral protocerebrum these neurons project to and how the pheromone neural circuit at this level processes information with opposite valences.

The male moth studied here, *Helicoverpa armigera* (Lepidoptera, *Noctuidae*, *Heliothinae*), utilizes *cis*-11-hexadecenal (Z11-16:Al) as the primary pheromone component and *cis*-9-hexadecenal (Z9-16:Al) as a secondary component (*Kehat and Dunkelblum, 1990*), in a species-specific ratio ranging from 100:2.5 to 100:4.5 (*Berg et al., 2014*; *Hillier and Baker, 2016*). Notably, the secondary constituent is identical with the primary pheromone component of the coresidential and closely related species, *Helicoverpa assulta*, in which Z11-16:Al and Z9-16:Al form the pheromone blend in a range of ratios from 5.7:100 to 6.7:100 (*Berg et al., 2014*; *Hillier and Baker, 2016*; *Wang et al., 2005*), and could thus act as an aversive signal at high concentrations. Another female-produced minor component, *cis*-9-tetradecenal (Z9-14:Al), undoubtedly plays a dual role in *H. armigera*. At higher dosages, it acts as a behavioral antagonist inhibiting the pheromone attraction, that is, when more than five units of Z9-14:Al are added to 100 units of the specific binary pheromone blend (*Gothilf et al., 1978*; *Kehat and Dunkelblum, 1990*). However, at lower dosages, it acts as an agonist, that is, when 0.3–5 units are added to 100 units of the specific binary pheromone blend (*Wu et al., 2015*; *Zhang et al., 2012*). This functional duality of a single molecular component indicates the complexity of the neural circuits processing pheromone information. The system must be capable of encoding attractive and aversive signals appropriately in order to elicit coordinated responses maximizing reproductive fitness.

In the study presented here, we characterized the male-specific PNs passing along the prominent mALT and the slightly thinner mlALT in *H. armigera*, focusing particularly on their projection patterns in the lateral protocerebrum. By applying the intracellular recording/staining technique, combined with calcium imaging experiments, we have mapped the projection patterns of physiologically identified MGC neurons. Based on the detailed neuronal architecture of individual PNs and their high-resolution spiking data we computed a descriptive framework reflecting the activity pattern dynamics within distinct protocerebral subregions induced by four behaviorally relevant female-produced odorants. The current study provides solid evidence for a spatial arrangement within the relevant protocerebral neuropils of the male moth demonstrating distinct regions receiving input about separate or intermixed female-released signals associated with attraction versus inhibition of attraction.

## Results

### Mapping odor-evoked responses of the MGC output neurons by means of calcium imaging

The projection pattern of the male-specific *sensory* neurons onto the MGC units was previously mapped via bath application calcium imaging studies (*Kuebler et al., 2012*; *Wu et al., 2013*; *Wu et al., 2015*). To measure the output signals from the same three MGC glomeruli, we performed calcium imaging measurements of odor-evoked responses in a group of PNs exclusively. By applying a calcium-sensitive dye (Fura-2) into the calyces (*Figure 2A*), we label primarily the population of medial-tract uniglomerular neurons (*Ian et al., 2016*; *Kymre et al., 2021*). In the species used here,

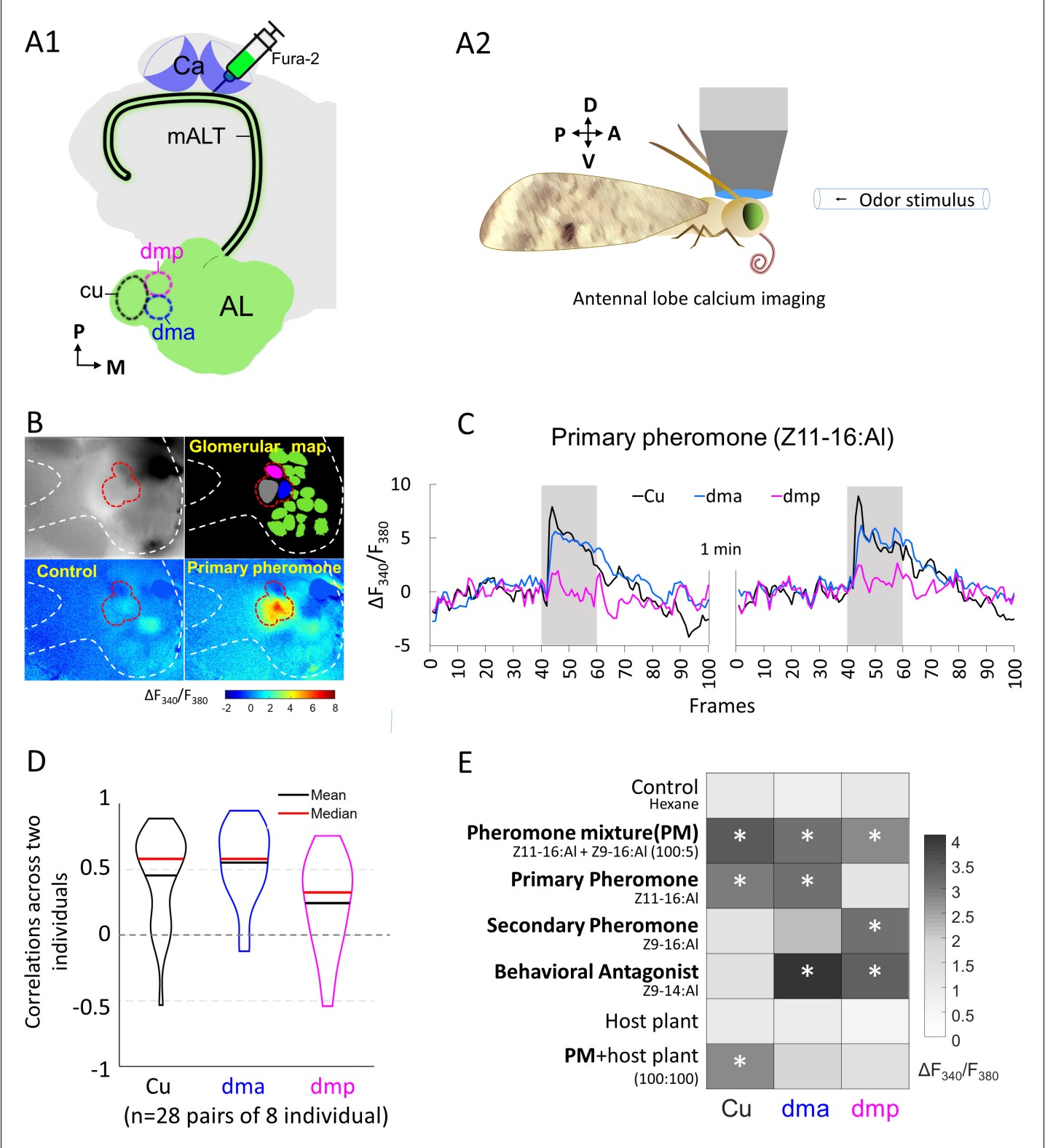

**Figure 2.** Macroglomerular complex (MGC) neurons confined to the medial antennal-lobe tract (mALT) and their odor responses during calcium imaging. (**A1**) Illustration of the retrograde staining with the calcium indicator (Fura-2) from calyx (Ca) labeling a population of the antennal lobe (AL) output neurons exclusively confined to the mALT, including the mALT MGC neurons. (**A2**) Schematic showing the placement of the moth during calcium imaging providing dorsal orientation of the brain. (**B**) Characteristic examples of calcium imaging data on MGC: top left, image of an AL stained with Fura from the Ca; top right, a processed image showing a map of recognized glomeruli; down left and down right, heat maps of
*Figure 2 continued on next page*

*Figure 2 continued*

responses to the control and primary pheromone, respectively. Dashed *white* and *red* lines mark the AL and the MGC region, respectively. (C) Calcium signal rises in response to the primary pheromone in neurons innervating distinct MGC units. An average baseline activity, that is, the Fura signal representing the ratio between 340 and 380 nm excitation light ($F_{340}/F_{380}$) from 0.5 to 2.5 s (frames 5–25, within 4 s spontaneous activity) was selected and set to zero. The trace of neuronal activity, specified as $\Delta F_{340}/F_{380}$, illustrates the changes in fluorescent level in two repeated stimulations with 100 ms sampling frequency. The interval between stimulations was 1 min. Gray bar, the duration of the stimulation period (2 s). (D) Violin plot of consistent tests across eight individuals. (E) Mean response amplitudes of a population of PNs innervating the same MGC units to all presented stimuli (n = 8), where * indicates a significant response compared with control. Box plots of the response amplitudes are shown in *Figure 2—figure supplement 1*. Cu, cumulus; dma, dorsomedial anterior unit; dmp, dorsomedial posterior unit.

The online version of this article includes the following figure supplement(s) for figure 2:

**Figure supplement 1.** Mean response traces (n = 8 moths) of the macroglomerular complex (MGC) units during stimulation with all presented stimuli (*top*), where * represents a statistically significant deviation from the Fura signal evoked by control (with 95% confidence level).

**Figure supplement 2.** Cross-stimuli correlation plot of mean response traces of the macroglomerular complex (MGC) units during application of all presented stimuli (n = 8 moths).

*H. armigera*, the MGC comprises three units (*Skiri et al., 2005*; *Zhao et al., 2016*) receiving input from three OSN categories: (1) the cumulus from OSNs tuned to the primary component, Z11-16:Al, (2) the dorsomedial posterior (dmp) unit from OSNs tuned mainly to the secondary component, Z9-16:Al, and (3) the dorsomedial anterior (dma) unit from OSNs tuned to the behavioral antagonist/enhancer, Z9-14:AL (*Wu et al., 2015*). For simplicity, Z9-14:Al is mentioned as a behavioral antagonist in the subsequent text. The imaging data of mALT PNs connected to each of the three MGC units was obtained from dorsally oriented brains (*Figure 2B*). One example of repeated traces illustrates that the increase in intracellular $Ca^{2+}$ in the MGC during antennal stimulation with the two pheromone components was steady (*Figure 2C*).

We checked whether the population of mALT PNs from each MGC unit showed consistent responses across individual insects. The response consistency of a group of PNs can be quantified as the Pearson's correlation coefficient of the response vectors in two individual insects, where each vector contains the trial-averaged responses of an individual to a given set of odors (*Mittal et al., 2020*; *Schaffer et al., 2018*). The average correlation between the cumulus medial-tract PNs responses (mean calcium signal during stimulation windows subtracted by that during the pre-stimulation period) across individuals was 0.45. The corresponding average correlations of PNs from the dma and dmp units were 0.55 and 0.24, respectively (*Figure 2D*). These paired-individual correlations were greater than the nonlinear relationship with a chance level of 0 (*t*-test, p's<0.002), confirming the general response consistency across different insects. We then profiled the pheromone responses in mALT PNs from each MGC unit in eight insects, by comparing the Fura signals evoked by each stimulus during the 2 s stimulation window with the control (hexane; *Figure 2E*). The average calcium traces and response amplitudes observed in these insects are presented in *Figure 2—figure supplement 1*. As expected, the cumulus PN population showed a pronounced activation during stimulation with the primary pheromone and the pheromone blend. The dma output neuron population, in turn, responded not only to the behavioral antagonist but also to the primary pheromone and the pheromone blend. The dmp PNs responded to the secondary pheromone and the behavioral antagonist, as well as to the pheromone mixture. All these responses showed a phasic component that decayed over the course of the stimulation period. Taken together, each stimulus with different behavioral relevance evoked a unique activation pattern in the three MGC units (*Figure 2E*). We also analyzed the mean calcium traces of eight individuals across seven stimuli (see *Odor stimulation* section in Materials and methods). The across-stimuli correlation plot illustrates that, unlike the relatively defined responses of the PNs innervating the cumulus and the dmp unit, the population of dma PNs evoked a broad activation pattern including responses to all female-produced chemicals tested (*Figure 2—figure supplement 2*). In contrast to the previously reported narrow tuning of the male-specific OSNs (*Wu et al., 2015*), we found that the odor response profiles of the medial-tract output neurons were considerably more intricate.

## Identification of individual MGC projection neurons and their odor representation in protocerebral neuropils

We next aimed to elucidate the functioning and morphology of individual neurons involved in processing pheromone information. Intracellular recording and staining were executed from the thick dendrites of MGC output neurons, including PNs confined to both the mALT and mlALT. We recorded 32 PNs across 32 preparations (*Figure 3A–F*, *Figure 3—figure supplements 1–2* and *Videos 1–10*). Among them, 29 PNs were confined to the prominent mALT and three to the substantially thinner mlALT (the morphological features of individual neurons are summarized in *Figure 3—source data 1*). The main types of these MGC-PNs can be accessed via the Insect brain database (*InsectbrainDB, 2021*).

Generally, the uniglomerular medial-tract MGC-PNs projected sparsely to a restricted area in the inner layer of the calycal cups before targeting at least one of four regions in the lateral protocerebrum, that is, the ventrolateral protocerebrum (VLP), the superior lateral protocerebrum (SLP), the anteroventral lateral horn (LH), and the posterodorsal part of the superior intermediate protocerebrum (SIP) (*Figure 3A–D*). The protocerebral projection patterns of mALT PNs innervating the same MGC unit(s) were homogeneous. The three MGC-PNs following the mediolateral ALT were multiglomerular and projected directly to the lateral protocerebrum without innervating the calyces (*Figure 3A,E–F*). In total, the most frequently recorded neuron type was the uniglomerular medial-tract PN originating in the cumulus and targeting the VLP and SLP (*Figure 3—figure supplement 1G*).

We found that the connectivity between the MGC units and the targeted protocerebral neuropils of each PN displayed an interesting pattern (*Figure 3—figure supplement 1H*), where different neuropils served as output regions for distinct groups of MGC-PN types: (1) the VLP for all MGC output neurons, (2) the SLP and SIP for PNs innervating cumulus, and (3) the LH for PNs originating from each of the two smaller MGC units as well as multiglomerular medial-tract PNs arborizing in the whole MGC. Based on the morphological and electrophysiological data from each individual MGC-PN, we generated a descriptive framework mapping the neural activity patterns within the relevant protocerebral areas. Here, the mean firing traces of recorded MGC-PNs with projections into the same individual output region were computed. The response amplitude of the simulated mean trace was used as the simulated value (*Figure 3G*). The primary pheromone signals were processed within the SLP and SIP whereas the minor components were represented in the LH. It appeared that odor signals evoking attractive and aversive behaviors would excite different protocerebral subregions regardless of the neurons' glomerular arborizations. The VLP, in contrast, processed signals from all components.

Furthermore, to quantify how valid our description was, we simulated neuronal responses to each female-produced odor with respect to four types of projection combinations, that is, (1) VLP + SLP + SIP, (2) SLP + VLP, (3) VLP alone, and (4) VLP + LH. The simulation was based on average traces of described neural activities from different output regions shown in *Figure 3G*. The simulated value was then compared with the mean of recorded neuronal activity of the neurons having the same projection (*Figure 3J*). The simulation was consistent with the recorded data, demonstrated by strong correlations between the simulated and recorded neural activities across the four different odors ($R^2 > 0.6$, *Figure 3H2*).

## Output regions of PNs across distinct ALTs

Previous and present data from *H. armigera*, including confocal images and 3D reconstructions of MGC-PNs confined to the medial, mediolateral, and lateral ALTs, indicated that their output regions could potentially overlap in specific protocerebral regions (*Chu et al., 2020a*). To explore whether such PNs intersect, we next performed multi-staining experiments. One preparation contained three individual MGC neurons confined to each of the three main tracts (*Figure 4A*). Here, one mALT PN and one lALT PN were strongly labeled, whereas the mlALT PN was moderately stained. The lateral-tract neuron projected to the ipsilateral column of the SIP, as previously described by *Chu et al., 2020a*. This particular region was not innervated by any other MGC-PN types. PNs confined to the medial and mediolateral tract, however, overlapped in the VLP (dashed circle in *Figure 4A*). Two co-stained cumulus-PNs in another preparation, included one lateral-tract PN targeting the anteriorly

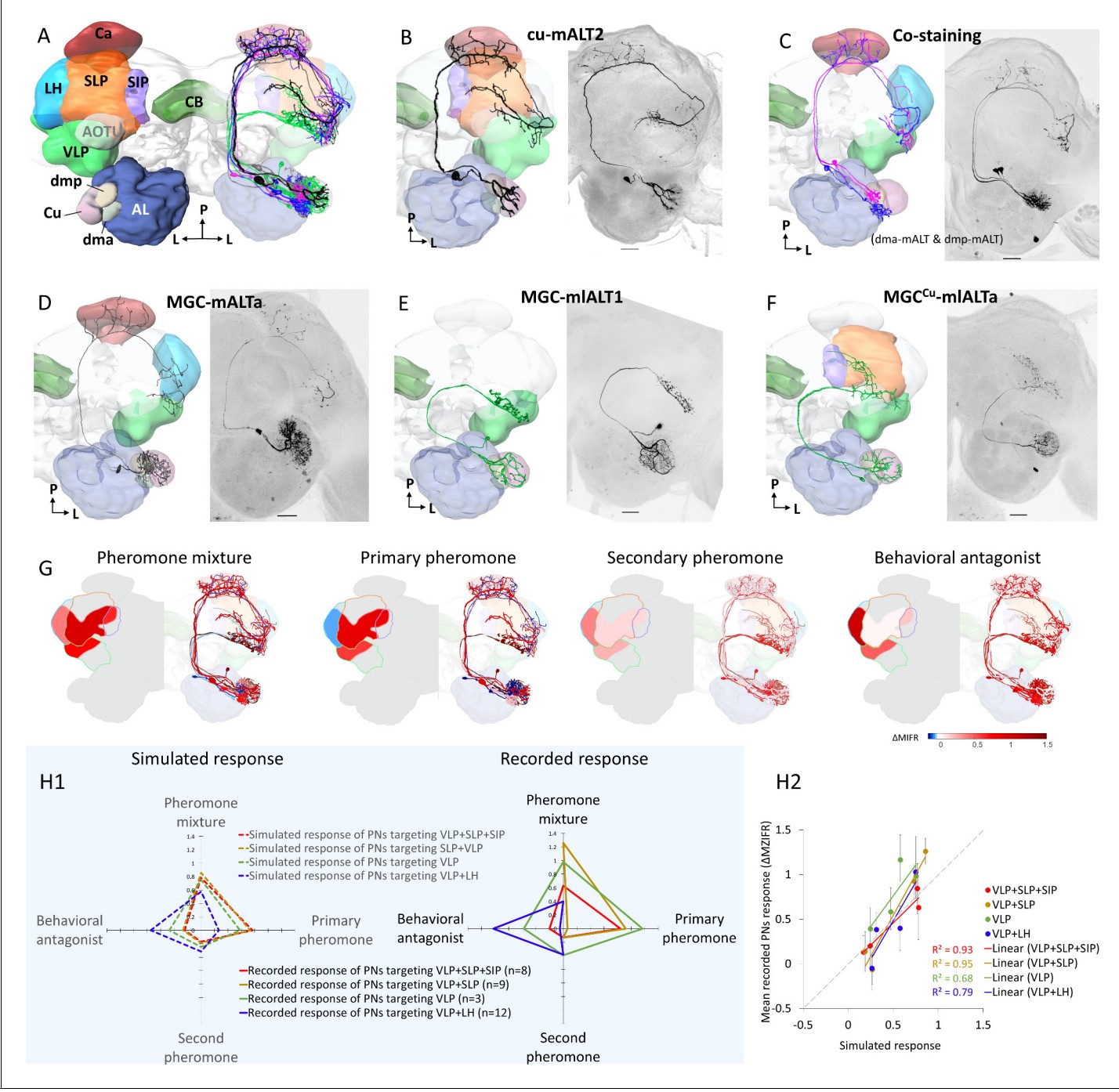

**Figure 3.** Morphological features of physiologically identified projection neurons (PNs) in the medial and mediolateral tracts – reconstructions, confocal images, and descriptive framework. (**A**) Diagram of the brain neuropils targeted by the macroglomerular complex (MGC) output neurons in a dorsal view. Color codes are in correspondence with all other figure panels. AL, antennal lobe; Ca, calyces; CB, central body; LH, lateral horn; SIP, superior intermediate protocerebrum; SLP, superior lateral protocerebrum; VLP, ventrolateral protocerebrum. (**B**) A typical example of uniglomerular PN in the medial antennal-lobe tract (mALT), originating from cumulus. (**C**) Two co-labeled mALT PNs targeting the same protocerebral subregions, one innervating the dma (blue) and the other the dmp (magenta). (**D**) A multiglomerular medial-tract PN with dendritic branches innervating the MGC units homogeneously. (**E**) A mediolateral antennal-lobe tract (mlALT) PN with homogeneously distributed dendrites across all MGC units. (**F**) Another type of mlALT PN, with dendrites in all MGC units, but dense arborizations only in the cumulus. All neurons in (**B–F**), which were labeled via dye injection after intracellular recording, were 3D-reconstructed and manually registered into the representative brain by means of the AMIRA software, based on the reconstructed neuropils innervated by the neuron of interest. Unique neuron IDs are indicated in (**B, D–F**). D, dorsal; L, lateral; M, medial; P, posterior. Red asterisks indicate weakly co-labeled neurons. Scale bars: 50 µm. (**G**) Descriptive framework showing stimulus-evoked neural activation within

*Figure 3 continued on next page*

*Figure 3 continued*

marked areas of the relevant neuropils. The outlines are based on the computed average firing traces across recorded MGC-PNs with projections into the same subarea in the individual output neuropil (for details of PN assembling in each neuropil see *Figure 3—figure supplement 1H*). In each subpanel, the heat map reflecting the response amplitude of the computed firing trace within the relevant neuropil is shown in the hemisphere to the left, while the mean responses of each PN type (presented in B–F) are shown in the hemisphere to the right. (H) Comparison between simulated neuronal responses and actual recorded neuronal responses to each female-produced odor. The simulations (left in H1) were based on described value showed in (G). The scatter plot in (H2) visualized the correlation between simulated and recorded data across four stimuli.

The online version of this article includes the following source data and figure supplement(s) for figure 3:

**Source data 1.** Overview of individual projection neuron morphologies.

**Figure supplement 1.** 3D reconstructed macroglomerular complex projection neurons(MGC-PNs), confined to the medial antennal-lobe tract (mALT) and mediolateral antennal-lobe tract (mlALT), including reconstruction of the neuropils innervated by the neuron of interest.

**Figure supplement 2.** Morphology of all individually recorded macroglomerular complex projection neurons(MGC-PNs).

---

located column of the SIP and one medial-tract PN targeting the VLP, SLP, and posterior SIP (*Figure 4B*). These PNs had no overlapping terminals.

## Physiological characteristics of the MGC-PNs

We characterized the electrophysiological features of the labelled MGC-PNs more in detail. First, we generated a heat map of every neuron's mean Z-scored instantaneous firing rate specifying the temporal response patterns to the four female-produced stimuli (*Figure 5*). In addition, we report, for each PN, responses that were significantly different from the prestimulation firing rate (*Figure 5—figure supplement 1*). For all PNs, examples of spike data and mean electrophysiological traces across repeated trials are shown in *Figure 5—figure supplements 2–7*. Each neuron was named with a unique ID. For PNs having dendritic innervations only in one of the MGC units, the neuron ID is expressed as 'innervated glomerulus-ALT number/letter,' and for PNs innervating multiple MGC units as 'MGC$^{main\ innervated\ glomerulus}$-ALT number/letter.' Here, the field of 'number/letter' represents two different concentration protocols (see 'Odor stimulation' section in the 'Materials and methods'). Finally, we averaged the responses of all individually recorded PNs and described how the sampled MGC-PNs represent pheromone signals across distinct MGC units. These data was then compared with the calcium imaging results obtained from populations of corresponding neurons (*Figure 6*).

## Medial-tract PN response-profiles were only partly congruent with OSN inputs

The physiological profiles within each neuronal group innervating the same MGC unit varied to some extent, both with respect to odor discrimination and temporal response characteristics. Except for uniglomerular cumulus mALT PNs, the response patterns of MGC output neurons appeared rather similar in both high and low concentrations. The cumulus PNs commonly responded with increased spiking frequency to both low and high concentrations of the primary pheromone and the pheromone mixture (*Figure 5* and *Figure 5—figure supplements 2–3*). The phasic onset was most prominent for PNs in the low concentration protocol (*Figure 5—figure supplement 2D*); these neurons

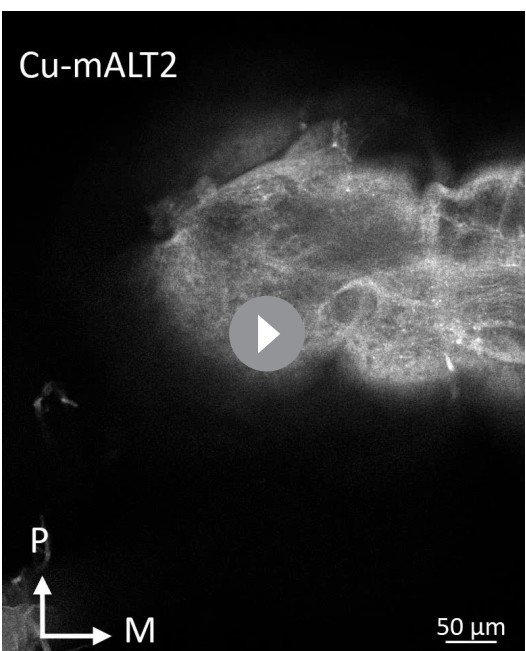

**Video 1.** Confocal stacks of neuron Cu-mALT2. The uniglomerular medial-tract neuron innervating the cumulus projects to the calyces, the ventrolateral protocerebrum (VLP), the superior lateral protocerebrum (SLP), and the superior intermediate protocerebrum (SIP).

https://elifesciences.org/articles/65683#video1

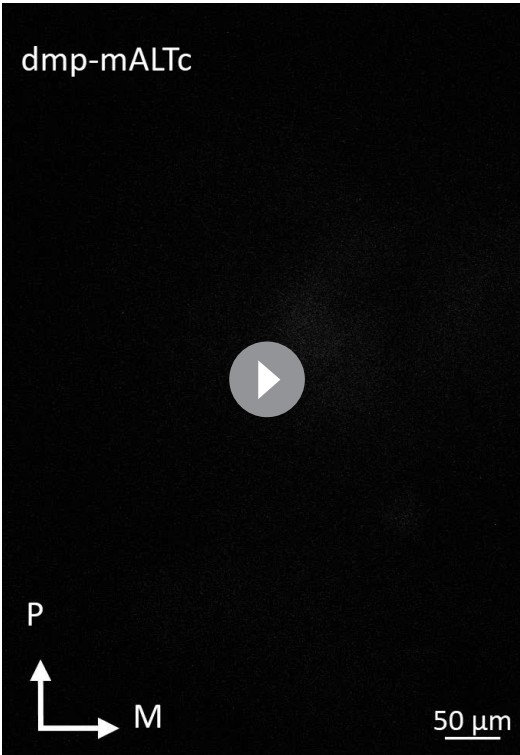

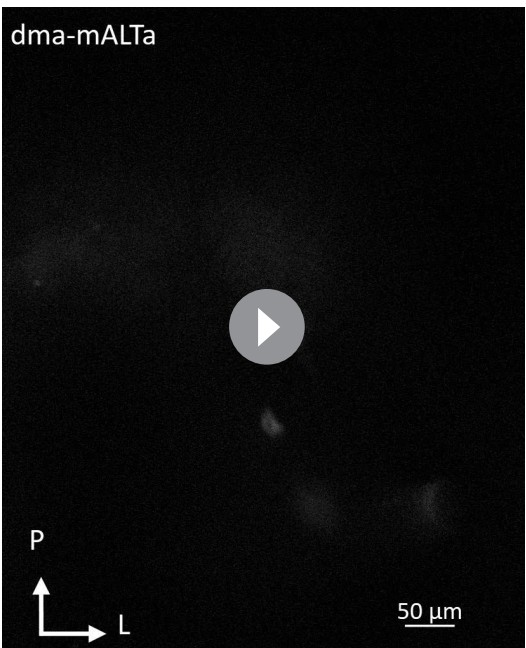

**Video 2.** Confocal stacks of neuron dmp-mALTc. The uniglomerular medial-tract neuron innervating the dmp unit projects to the calyces, the ventrolateral protocerebrum (VLP), and the lateral horn (LH).
https://elifesciences.org/articles/65683#video2

**Video 3.** Confocal stacks of neuron dma-mALTa. The uniglomerular medial-tract neuron innervating the dma unit projects to the calyces, the ventrolateral protocerebrum (VLP) and the lateral horn (LH), with extended terminals approaching the superior lateral protocerebrum (SLP).
https://elifesciences.org/articles/65683#video3

also appeared to discriminate between odorants more precisely than PNs in the high concentration protocol (*Figure 5—figure supplement 2E*).

This coincides with previous studies reporting that OSNs display less specific response profiles with increasing odor concentrations (*Malnic et al., 1999*; *Sato et al., 1994*). Notably, a part of the cumulus-PNs had atypical responding patterns (*Figure 5—figure supplement 3*). The quantification of the *onset* and *peak* response latency illustrated a delayed *peak* in response to the secondary pheromone (*Figure 5—figure supplement 3G*). Therefore, these responses might be mediated by local computation involving local interneurons or multiglomerular MGC-PNs.

The responding profile of the uniglomerular dma PNs seemed somewhat complicated but were in line with their projection patterns: PNs having fine branches approaching the SLP in addition to the terminals in the VLP and LH, were excited by the behavioral antagonist and the secondary pheromone (*Figure 5—figure supplement 4A–C*), while PNs with restricted projections in the VLP and LH were not (*Figure 5—figure supplement 4D–F*). In addition, the multiglomerular PN arborizing in the dma and three posterior-complex glomeruli showed no response to any stimulus (*Figure 5—figure supplement 4G–K*). The four uniglomerular mALT PNs innervating the dmp were stimulated with the high-concentration protocol only. Even though the dmp unit reportedly receives input about the secondary pheromone and the behavioral antagonist (*Wu et al., 2015*), only one of the dmp-PNs identified here was significantly excited by both of these components (*Figure 5—figure supplement 5*). Of the three remaining dmp-PNs, one was excited by the behavioral antagonist exclusively, whereas two displayed no responses during any stimulus application.

The AL innervations of the four multiglomerular medial-tract PNs innervating all MGC units varied considerably. To investigate a putative association between MGC innervation and response characteristics, we quantified the dendritic density of each PN by measuring the fluorescence intensities within the separate MGC units (*Figure 5—figure supplement 6*), as previously performed (see *Chu et al., 2020b*; *Kc et al., 2020*). Most of these PNs having terminal projections in the VLP and

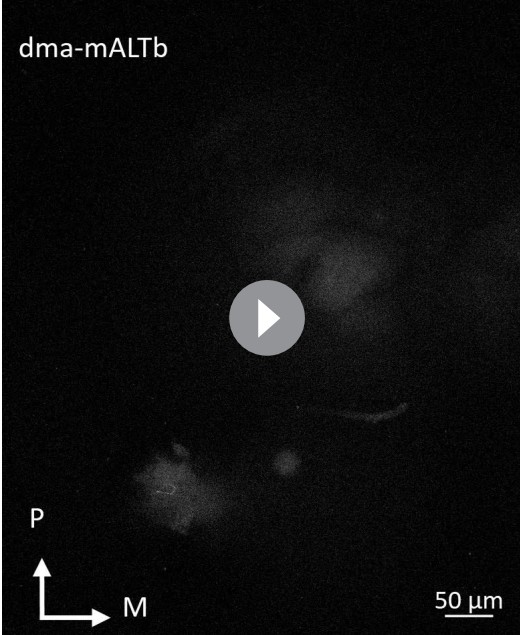

**Video 4.** Confocal stacks of neuron dma-mALTb. The uniglomerular medial-tract neuron innervating the dma unit projects to the calyces, the ventrolateral protocerebrum (VLP) and the lateral horn (LH). https://elifesciences.org/articles/65683#video4

## Representation of output signals from MGC-PNs

Based on the precise electrophysiological and morphological data obtained from the intracellular recordings, we took the mean response of MGC medial- and mediolateral-tract PN types, to provide a summary of odor-stimuli representation across the separate MGC units (*Figure 6A*). We first categorized the neurons into four groups based on their dendritic arborization, that is, in the cumulus, dma, dmp, or in all MGC units. The mean responses within the different neuron groups showed that mALT PNs with dendrites in the cumulus deal with the primary pheromone, and those in the dmp unit with the secondary pheromone and the behavioral antagonist. This is consistent with the calcium imaging results on medial-tract PNs (*Figure 6B*), and with former reports on response properties of the corresponding OSNs (*Wu et al., 2015*). However, unlike previous reports on OSN input, both our individual PN recordings and calcium imaging tests indicated a role for the dma PNs in processing information not only about the behavioral antagonist, but also, surprisingly, about the primary pheromone and pheromone mixture. However, averaging

LH responded to the behavioral antagonist. This indicated that each PN's responding profile did not correspond precisely with its dendritic architecture, but rather with its output region.

## The responses of mlALT PNs corresponded with OSN inputs

The three mediolateral-tract MGC-PNs consisted of two morphological sub-types displaying different response profiles. The singular PN constituting the first sub-type targeted the VLP, SLP, and SIP, while its dendrites filled the cumulus densely and the dma, dmp, and some posterior complex glomeruli sparsely. This neuron responded with a weak and early phasic excitation to most stimuli. Yet, the primary pheromone induced the only significant response (*Figure 5*; *Figure 5—figure supplement 7*). The other sub-type, including two mlALT PNs, projecting to the VLP exclusively, covered the MGC units quite uniformly. These PNs were more broadly tuned by responding with excitation to all stimuli. However, their responses to the pheromone mixture and the primary pheromone lasted substantially longer than the phasic excitations elicited by the secondary component and the behavioral antagonist (*Figure 5*; *Figure 5—figure supplement 7*).

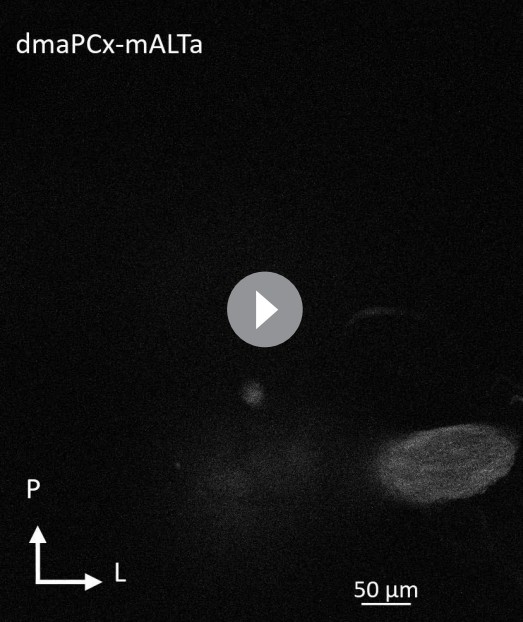

**Video 5.** Confocal stacks of neuron dmaPCx-mALTa. The multiglandular medial-tract neuron innervating the dma unit and three posterior complex glomeruli projects to the calyces, the ventrolateral protocerebrum (VLP), and the lateral horn (LH). https://elifesciences.org/articles/65683#video5

the activity across the uniglomerular dma neurons demonstrated that the most potent stimuli for these PNs were the behavioral antagonist and the pheromone mixture, while the primary pheromone elicited only minor increases in mean firing rates. Here, the influence of multiglomerular MGC-PNs may affect the responses to the mixture and the primary pheromone. For the fourth group, which consisted of the multiglomerular mALT and mlALT PNs arborizing in the entire MGC, the averaged data demonstrated activation by all female-produced components. Next, we compared the electrophysiological data of individual PNs with the calcium imaging results of populations of PNs (*Figure 6B*). As the calcium imaging experiment could not distinctly measure the responses from the subpopulation of uniglomerular PNs, the more broadly tuned responding patterns were probably a result of signals from both multi- and uniglomerular MGC-PNs.

## Discussion

In moths, the sex pheromone is usually produced as a blend of several components in a species-specific ratio (*Baker and Hansson, 2016*; *Christensen et al., 1995*). Principally, the male is attracted by the major pheromone component released by a conspecific female. While minor components do not elicit upwind flight on their own, they may enhance attraction (*Kehat and Dunkelblum, 1990*), and can also serve as behavioral antagonists, since such components are often produced by heterospecific females (reviewed by *Berg et al., 2014*) or immature conspecific females (*Chang et al., 2017*). Despite these innate responses, the mate-searching activities, including an initial surge and zig-zag casting behavior (*Cardé and Willis, 2008*; *Kuenen and Cardé, 1994*; *Vickers and Baker, 1994*), are not simple olfactory reflexes. The data presented here, comprising a large number of MGC medial-tract PNs mainly originating from one of three easily identifiable glomeruli, indicated that such behavioral responses are related not only to spatial representation of odor valence in the lateral protocerebrum, but also to the intensity of the relevant signals. Taken together, these results confirm our previous suggestions signifying that the lateral protocerebrum contributes to innate attraction/avoidance behavior based on composite odor input coding (*Chu et al., 2020a*).

### Pheromone signaling along parallel ALTs

Parallel processing of AL output neurons has been reported in many insects (*Figure 1B*, reviewed by *Galizia and Rössler, 2010*). Similar

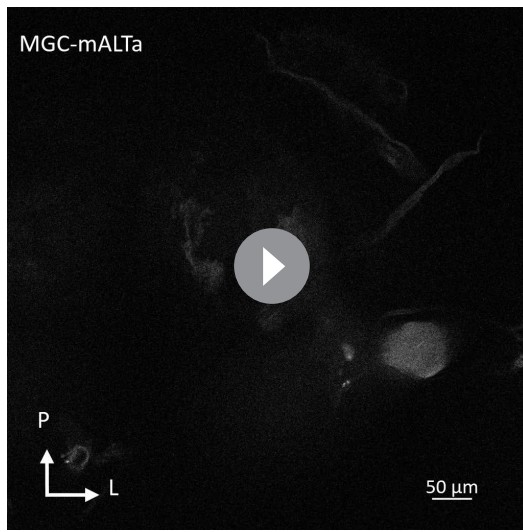

**Video 6.** Confocal stacks of neuron MGC-mALTa. The multiglandular medial-tract neuron has dendrites evenly distributed across three macroglomerular complex (MGC) units. It projects to the calyces, the ventrolateral protocerebrum (VLP) and the lateral horn (LH).
https://elifesciences.org/articles/65683#video6

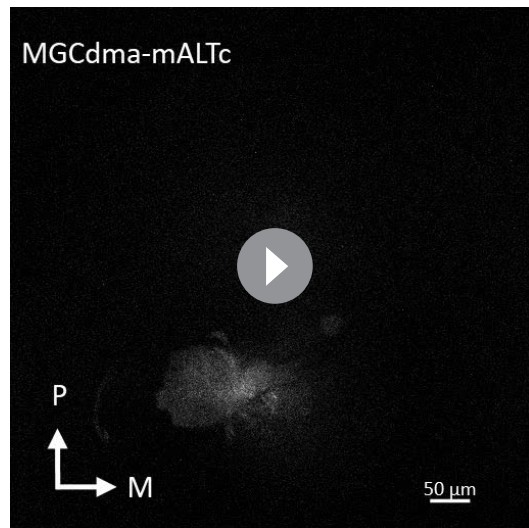

**Video 7.** Confocal stacks of neuron MGC$^{dma}$-mALTc. The multiglandular medial-tract neuron innervates the dma densely, while the cumulus and dmp are sparsely innervated. It projects to the calyces and the ventrolateral protocerebrum (VLP).
https://elifesciences.org/articles/65683#video7

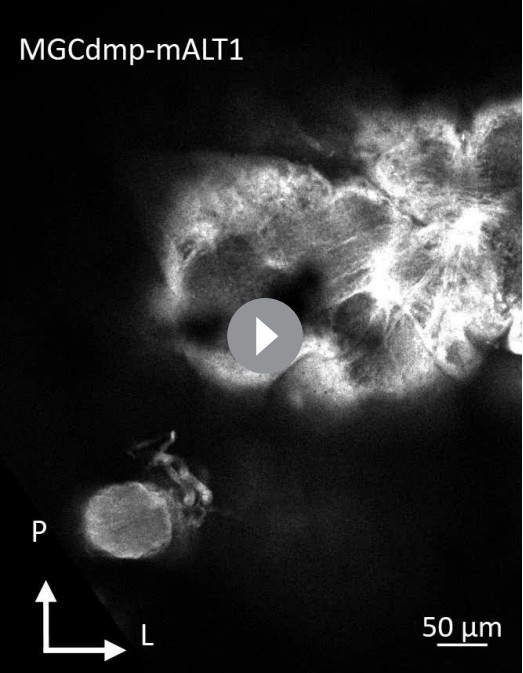

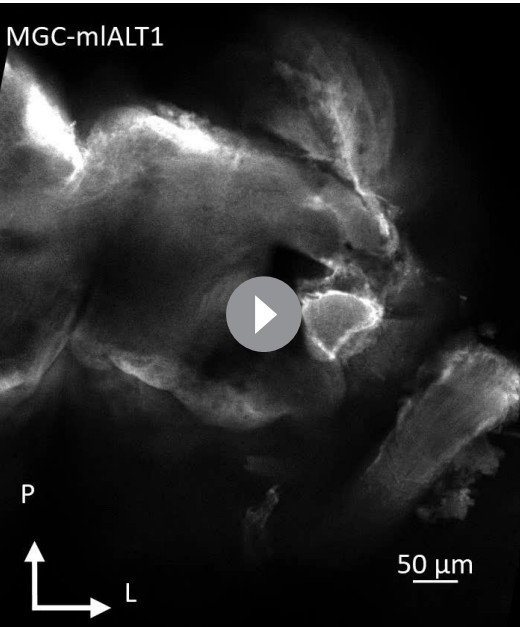

**Video 8.** Confocal stacks of neuron MGC^dmp-mALT1. The multiglandular medial-tract neuron innervates the dmp densely, while the cumulus and dma are sparsely innervated. It projects to the calyces, the ventrolateral protocerebrum (VLP), and the lateral horn (LH). https://elifesciences.org/articles/65683#video8

**Video 9.** Confocal stacks of neuron MGC-mlALT1. The multiglandular mediolateral-tract neuron innervating the macroglomerular complex (MGC) units evenly projects to the ventrolateral protocerebrum (VLP) with blebby terminals. https://elifesciences.org/articles/65683#video9

to the most studied insect model, the fruit fly *Drosophila melanogaster*, the moth also has three main tracts connecting the lateral protocerebrum and the AL. Male-specific signals originating from the MGC of the moth is carried along these tracts. The findings presented here, comprising high-resolution morphological and physiological data on MGC-PNs passing along the medial and mediolateral ALT in *H. armigera*, complement newly reported and corresponding results on MGC-PNs confined to the third tract, the lateral ALT, in the same species (*Chu et al., 2020a*). Interestingly, both medial- and lateral-tract MGC-PNs are mainly uniglomerular. However, the medial tract includes PNs originating from each of the three MGC-units whereas the lateral tract is reported to connect with the cumulus exclusively. Besides, their projection patterns are different; while medial-tract MGC-PNs extend widespread terminal branches into various protocerebral regions, including the SLP, posterior SIP, and VLP, the (unilateral) lateral-tract MGC-PNs converge within a restricted region anteriorly in the SIP called the column (*Chu et al., 2020a*). To clarify how the pheromone-information is carried along not only the medial and the lateral ALT but also in the somewhat thinner mediolateral ALT, we integrated the morphological findings from the previous study with our current results, and constructed a comprehensive map displaying the neural connections between MGC/AL glomeruli and the protocerebral target areas for each tract (*Figure 7A–B*). Up to now, we have the largest collection of MGC-PNs from one distinct species, including detailed morphological and physiological characterization of 42 individual neurons (32 from the current study, 10 from *Chu et al., 2020a*). This assembly covers every previously reported MGC neuron type in heliothine moths (*Berg et al., 1998*; *Christensen et al., 1991*; *Christensen et al., 1995*; *Ian et al., 2016*; *Lee et al., 2019*; *Vickers et al., 1998*; *Zhao and Berg, 2010*; *Zhao et al., 2014*). Here, we demonstrate that the proportion of sampled neurons originating from each of the MGC units is in high correspondence with the volume of the relevant MGC unit (*Figure 7D*). Thus, the likelihood of discovering a substantial amount of new MGC-PN types is rather small.

This encouraged us to propose a framework representing the temporal response properties of all recorded MGC-PNs in the relevant protocerebral neuropils. The mean firing traces of recorded

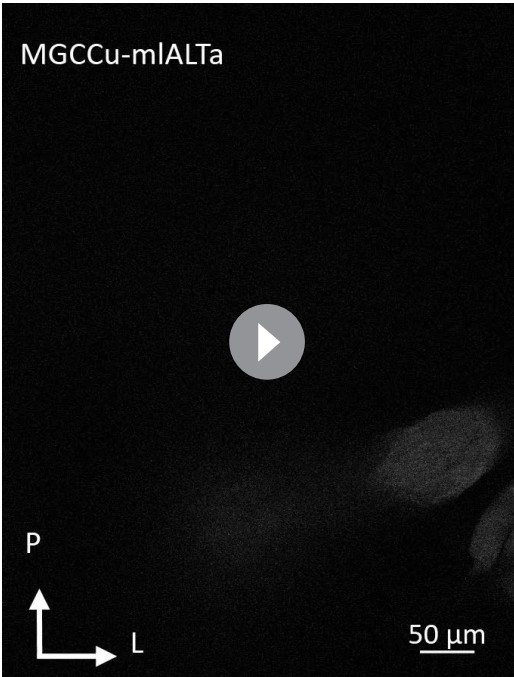

**Video 10.** Confocal stacks of neuron MGC[Cu]-mlALTa. The multiglandular mediolateral-tract neuron mainly innervating the cumulus has sparse dendrites in the dma and the dmp units. It projects to the ventrolateral protocerebrum (VLP), the superior lateral protocerebrum (SLP), and the superior intermediate protocerebrum (SIP).
https://elifesciences.org/articles/65683#video10

MGC-PNs with projections into the same individual output region were computed. As pheromone-signaling appeared to be particularly fast in the lateral-tract MGC-PNs (*Chu et al., 2020a*), we decided to plot the neurons' mean firing traces within the stimulation window instead of their mean response amplitudes. The 42 MGC-PNs included in this analysis, consisted of 29 neurons in the mALT, three in the mlALT, and 10 in the lALT (including four unilateral PNs and six bilateral PNs). Interestingly, when including the data from lateral-tract MGC-PNs (*Chu et al., 2020a*), we discovered a sequential and logic response pattern in the different protocerebral areas (*Figure 7B–C*). Here, the excitatory response to the pheromone mixture arose first in the SIP, as a typical phasic response. After 20 ms, a phasic-tonic response to the binary mixture appeared in the SLP and VLP. Finally, about 120 ms later, a weak signal lasting for 10 ms could be seen in the LH. The primary pheromone evoked similar temporal response patterns in the SIP, SLP, and VLP, but barely any kind of signal could be observed in the LH. The secondary pheromone elicited weak and delayed excitation in the VLP and LH, corresponding in time with the tiny peak evoked by the binary mixture in the LH. Finally, the behavioral antagonist induced a clear and long-lasting response in the LH and a considerably weaker activation in the VLP.

To examine the content validity of our framework, we investigated whether the presented outline onto the protocerebral neuropils would be fundamentally altered if additional data points were included. We first carried out a simple computational experiment in which we shuffled the firing traces of individual PNs, selecting a different assembly of neurons by randomly recruiting two-thirds of the data points in the group. The data shuffling was repeated five times, and each time a different assembly of neurons was included. The cross correlations between each of the data assemblies showed that neuronal response profiles were unchanged in the neuropils associated with different behavioral valences (*Figure 7F*). Then, to rule out the possibility that the high cross assemblies' correlation was resulting from individual PNs having comparable firing profiles, we computed the correlations of the firing traces between every two PNs (*Figure 7G*). It appeared that the firing traces of individual PNs were clearly less correlated in comparison with that across different shuffling assemblies.

Notably, the very fast response in the SIP region, which occurred during stimulation with the pheromone blend and the primary constituent exclusively, involves the previously reported lateral-tract MGC-PNs terminating in the column (*Chu et al., 2020a*). Comparing the response onset in the SIP with the other output regions, demonstrates that the lateral-tract PNs react prior to PNs in the other main tracts (*Figure 7C*). This is particularly interesting as the MGC neurons confined to medial tract in this species have a much lower spontaneous firing rate as compared with the lateral-tract PNs (*Chu et al., 2020a*). The 'quieter' background of the medial-tract PNs might facilitate registration of the pheromone signals. One possible explanation for this property could be that the medial-tract PNs to a large extent are wired with OSNs indirectly via local interneurons, while the lateral-tract PNs may have synaptic contact more directly with the OSNs. This coincides with the connectomical results shown for the sexually dimorphic glomerulus, VA1v, in the fruit fly, where the local interneurons have stronger connections with the mALT PNs than the lALT PNs. Besides, the number of presynaptic sites is much higher in the medial-tract PNs than in the lateral-tract PNs, and the

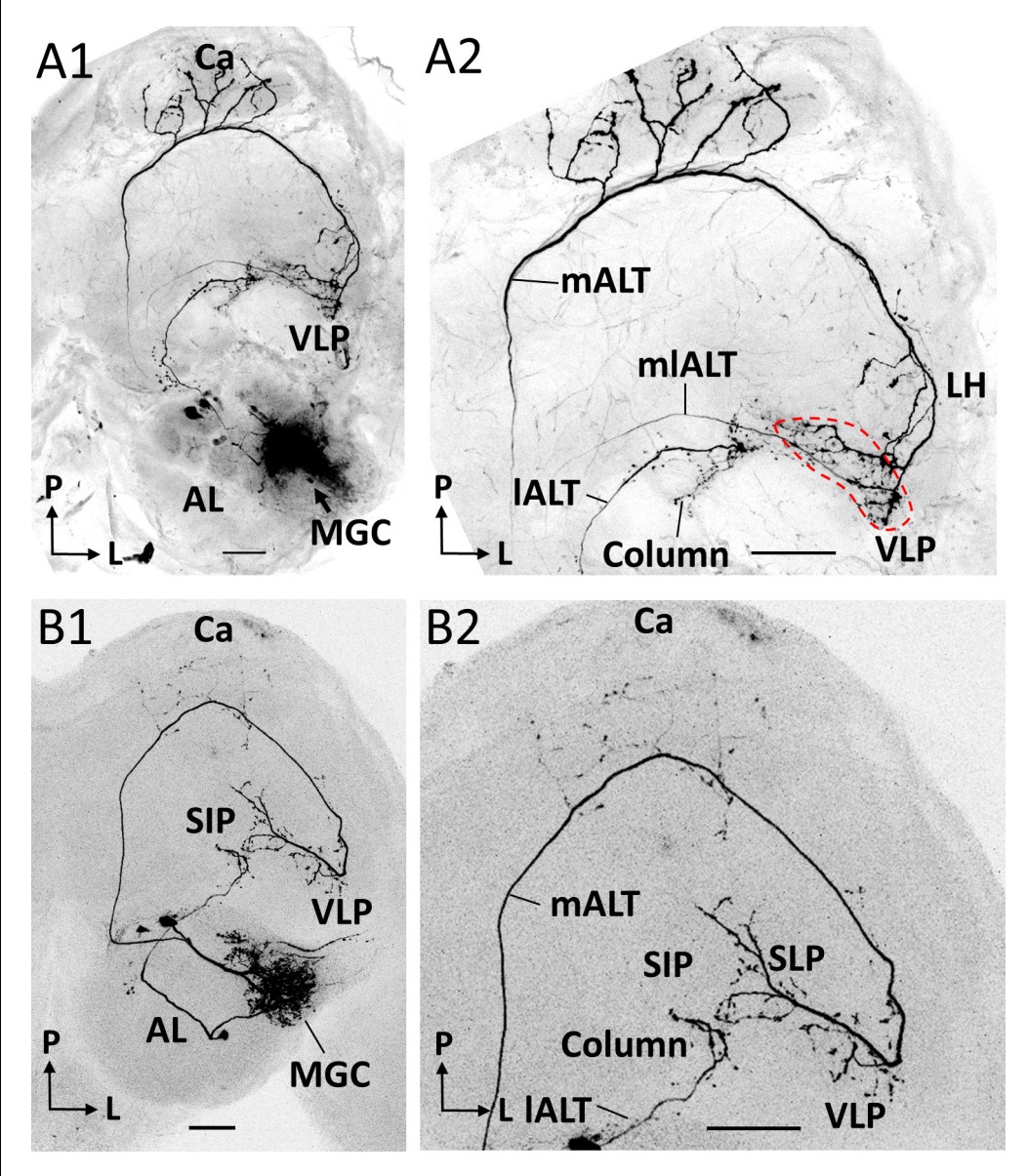

**Figure 4.** Co-labeled projection neurons (PNs) in distinct ALTs. (**A**) Application of dye in the macroglomerular complex (MGC) visualized three antennal-lobe projection neurons confined to the medial antennal-lobe tract (mALT), mediolateral antennal-lobe tract (mlALT), and lateral antennal-lobe tract (lALT), respectively (**A1**). The axonal terminals of the mlALT PN overlapped with the mALT PN in the VLP (*red* dashed lines in **A2**). (**B**) Two co-labeled cumulus PNs, confined to the mALT and lALT, respectively. The mALT PN has no overlap with the lALT PN. AL, antennal lobe; Ca, calyces; LH, lateral horn; SIP, superior intermediate protocerebrum; SLP, superior lateral protocerebrum; VLP, ventrolateral protocerebrum. L, lateral; P, posterior. Scale bars: 50 µm.

dendritic arborizations are also denser in the medial-tract PNs (*Horne et al., 2018*). As shown in the results, the MGC-PNs in the third and minor tract, the mediolateral ALT, had terminal projections partly overlapping with medial-tract MGC-PNs indicating that these tracts may be involved in activation of the same downstream neurons (*Figure 4*). This agrees with previous observations in *Manduca sexta* (*Homberg et al., 1988*). The lateral ALT, on the other hand, which innervates the column, seems to constitute a separate circuit. This reinforces our proposed concept implying that the mALT and mlALT are involved in fine-tuned odor coding, whereas the lALT plays a different role by initiating flight behavior evoked by the key component (*Chu et al., 2020a*; *Ian et al., 2016*).

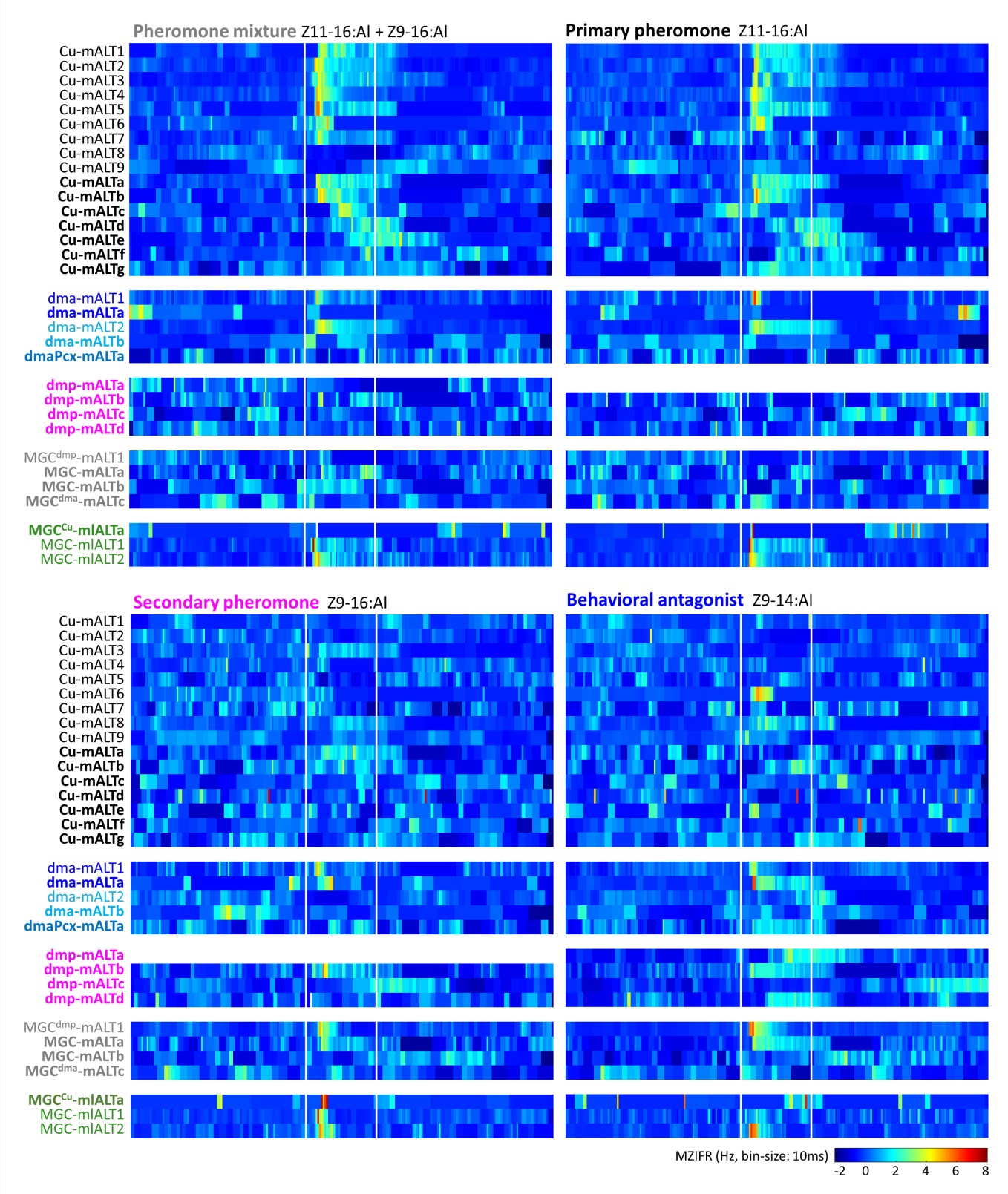

**Figure 5.** Temporal resolution index of spiking activity in the individual macroglomerular complex projection neurons (MGC -PNs). This plot displays the across-trials mean instantaneous firing rates (MZIFR) for each of the reported PNs in response to the four presented female-produced stimuli. The online version of this article includes the following figure supplement(s) for figure 5:

*Figure 5 continued on next page*

*Figure 5 continued*

**Figure supplement 1.** Physiological properties of individual macroglomerular complex projection neurons(MGC-PNs).

**Figure supplement 2.** The predominant response pattern of medial antennal-lobe tract (mALT) projection neurons (PNs) originating in the cumulus.

**Figure supplement 3.** Heterogeneous response profiles of cumulus-innervating medial antennal-lobe tract (mALT) projection neurons (PNs) with homogenous morphologies.

**Figure supplement 4.** Heterogeneous response profiles of two types of dma-innervating medial antennal-lobe tract (mALT) projection neurons (PNs).

**Figure supplement 5.** Response profiles of medial antennal-lobe tract (mALT) projection neurons (PNs) with uniglomerular dendrites in the dmp.

**Figure supplement 6.** Dendritic macroglomerular complex (MGC) innervation and physiological features of multiglomerular medial antennal-lobe tract (mALT) projection neurons (PNs).

**Figure supplement 7.** Response profiles of mediolateral antennal-lobe tract (mlALT) projection neurons (PNs) innervating the macroglomerular complex (MGC).

## Protocerebral output areas of MGC medial-tract neurons are organized according to behavioral valence

Despite the relatively widespread and partly overlapping terminal branches of the total assembly of medial-tract MGC-PNs, including 16 originating in the cumulus, four in the dmp, four in the dma, and five being multiglomerular, we discovered a pattern implying a spatial arrangement according to the cumulus-PNs versus the dma- and dmp-PNs. As discussed in detail below, this may correspond to the 'lateral-medial' representation of attraction- and aversion-related pheromone-signals previously reported in the protocerebrum of the closely related *Helicoverpa* species, *H. assulta,* and also in corresponding areas in the silk moth, *Bombyx mori* (*Kanzaki et al., 2003*; *Seki et al., 2005*; *Zhao et al., 2014*). In *H. armigera*, we found that signal representation related to attraction and inhibition is both segregated and integrated, but in distinct protocerebral neuropils.

### Inputs related to sexual attraction are primarily processed in the SLP and SIP

All uniglomerular mALT PNs innervating the SLP and SIP had their dendrites in the cumulus. These cumulus-PNs responded mainly to the primary pheromone. Although the behavioral antagonist excited 19% (3 of 16) of the cumulus-PNs as well, these responses were not at all comparable to those associated with the primary pheromone. All data, including our calcium imaging measurements (*Figure 2*) and mean intracellular responses (*Figure 6A*) confirm that the cumulus is devoted to process input about the primary pheromone. Notably, the multiglomerular mlALT PN innervating the SLP and SIP had its most dense innervations in the cumulus as well (*Figure 3F*). Thus, third-order

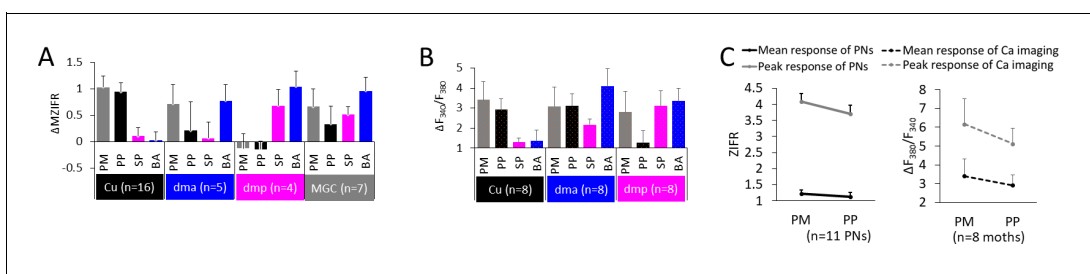

**Figure 6.** Summary of odor representation across macroglomerular complex (MGC) units. (**A**) Mean individual neurons' responses (ΔMZIFR; delta mean Z-scored instantaneous firing rate, for details see 'Spike data analysis' in 'Materials and methods') of MGC medial/mediolateral-tract projection neurons (PNs) during stimulation with distinct female-produced odors, sorted according to the dendritic arborizations. Data is presented in mean + sem. Note that the PNs listed as innervating the cumulus (Cu), dma, and dmp were uniglomerular, while the PNs in the MGC category were multiglomerular. (**B**) Mean calcium imaging responses (ΔF_{340}/F_{380}) of populations of MGC medial-tract PNs during stimulation with distinct female-produced odors, sorted according to the innervated glomerulus. The 'broad' responses in calcium imaging, as compared with the electrophysiological data, may be derived from the inclusion of multiglomerular mALT PNs. (**C**) Mean responses to pheromone mixture and primary pheromone in individual PNs (left) and in calcium imaging tests (right). Data is presented in mean + sem. PM, pheromone mixture; PP, primary pheromone; SP, secondary pheromone; BA, behavioral antagonist.

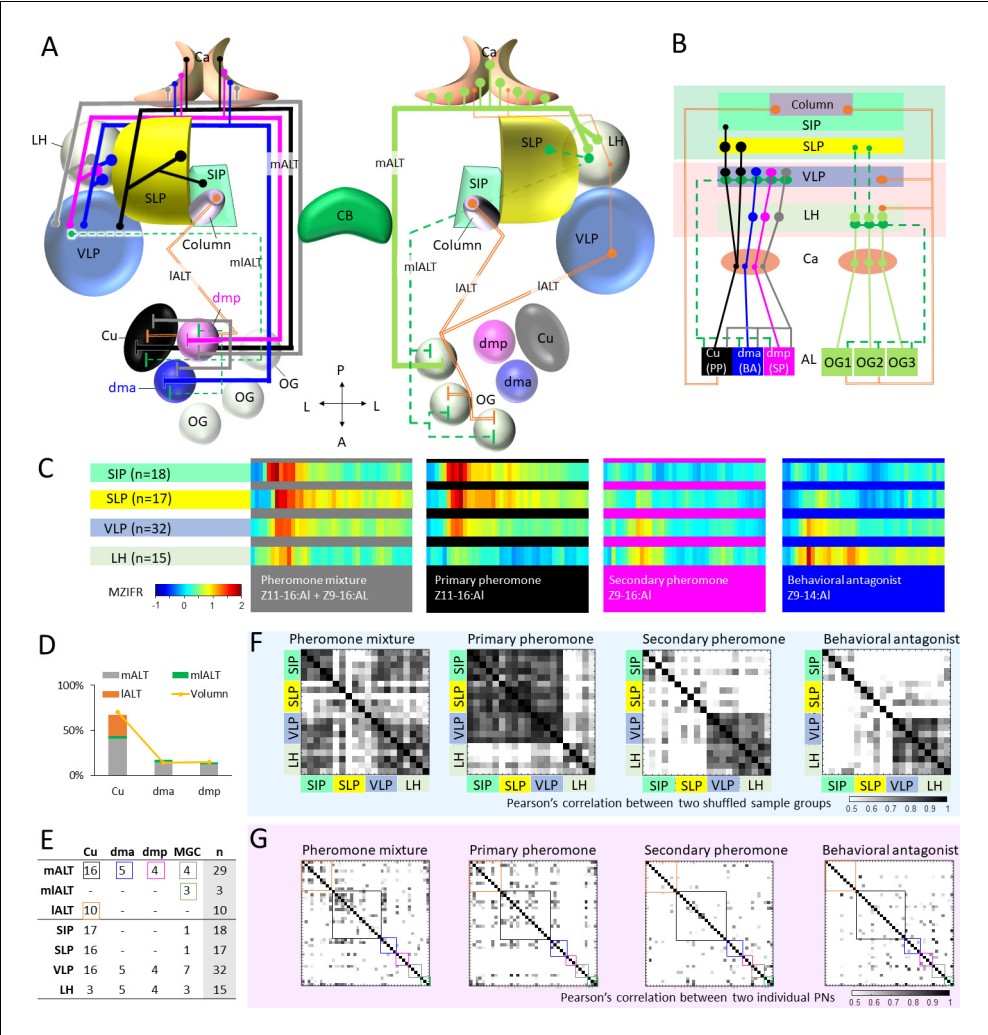

**Figure 7.** Summary of morphological and physiological features typifying macroglomerular complex projection neurons (MGC-PNs) across the three main tracts. (**A**) A graphic representation of the olfactory pathways in male moth brain. The morphological features of MGC-PNs is displayed on the hemisphere to the left. Multiglomerular medial-tract MGC-PNs (*gray*) and uniglomerular PNs arborizing in the dmp (*magenta*) or dma (*blue*) project to the calyces (Ca), anteroventral LH, and VLP, while the uniglomerular PNs with dendrites in the cumulus (Cu; *black*) target the Ca, VLP, SLP, and posterior SIP. The multiglomerular mediolateral-tract PNs (dashed *green* lines) target primarily the VLP, whereas the uniglomerular lateral-tract PNs innervating the cumulus run directly to the column in the anterior SIP (***Chu et al., 2020a***). In the hemisphere to the right, a plethora of different PN types arborizing in the ordinary glomeruli (OG) and innervating several lateral protocerebral neuropils are indicated (***Ian et al., 2016***; ***Kymre et al., 2021***). (**B**) Scheme illustrating projections of MGC-PNs versus OG-PNs in the higher brain regions. The three main antennal lobe tracts (ALTs) are illustrated: Solid line, medial antennal-lobe tract (mALT); dashed line, mediolateral ALT (mlALT); double line, lateral ALT (lALT). (**C**) An overview of temporal response properties implied by the mean firing rate during odor stimulation (MZIFR; mean Z-scored instantaneous firing rate) of the assembly of PNs projecting to the same neuropil. The PNs included in this framework are shown in (**E**). Note that these PNs were sampled from the antennal lobe, while the length of axons and action potential transmission rates have not been included, the actual timing of synaptic output onto the protocerebral neuropils is thereby not represented. (**D**) Correspondence between the proportion of sampled neurons originating from each of the MGC units and that of the volume of each MGC unit (volume data from ***Zhao et al., 2016***) shows that our sampling strategy for the proposed framework (**C**) reflects the composition of the actual MGC-PN population. (**E**) Summary of recorded and labelled PNs' morphologies across the three main tracts, including 10 lALT MGC-PNs from our previous study (***Chu et al., 2020a***). (**F**) Computational experiment with shuffled electrophysiological data indicating correlation between PN responses to attractive versus non-attractive stimuli and separated output areas. The data shuffling was repeated five times, and each time two-thirds of the PNs projecting to the same

*Figure 7 continued on next page*

*Figure 7 continued*
neuropil were randomly recruited into the data assembly. The cross correlations between each of the data assemblies showed that neuronal response profiles were consistent with the neuropils associated with different behavioral valences. (**G**) Cross-correlations of the firing traces between every two PNs. The neuron types are indicated by the color-coded boxes, in correspondence with (**E**). AL, antennal lobe; Ca, calyces; CB, central body; LH, lateral horn; OG, ordinary glomeruli; SIP, superior intermediate protocerebrum; SLP, superior lateral protocerebrum; VLP, ventrolateral protocerebrum. A, anterior; L, lateral; P, posterior.

pheromone-processing neurons in the SLP and SIP should primarily be involved in computations related to attraction.

## Anteroventral parts of the LH process dma and dmp input

In contrast to the SLP and SIP, being innervated by the medial-tract cumulus neurons, the anteroventral LH was the target for the PNs originating in one of the two smaller MGC units, the dma and dmp. Thereby, input about the primary pheromone is processed dorsomedially to input about the secondary component and the behavioral antagonist. This lateral-medial separation resembles the spatial pattern previously reported in the closely related species, *H. assulta*, where the primary pheromone is represented medially to the interspecific signal (*Zhao et al., 2014*). In the 'older' species *H. assulta*, however, the secondary pheromone is not processed together with the behavioral antagonist, but with the primary pheromone. Summarizing the response patterns of the MGC-PNs innervating the LH (*Figure 7C*), we found that the highest mean firing rate appeared during application of the behavioral antagonist. The secondary pheromone also induced increased spike frequency, but substantially weaker. As mentioned in the introduction, the behavioral antagonist, Z9-14:Al, reduces mating-associated behaviors as soon as its ratio vs. binary pheromone blend increases to 5:100 (*Kehat and Dunkelblum, 1990*; *Wu et al., 2015*), while it enhances such behaviors when the ratio is between 0.3:100 and 5:100 (*Wu et al., 2015*; *Zhang et al., 2012*). Indeed, a small amount of this component, Z9-14:Al, is released by *H. armigera* females (ratio of Z9-14:Al to Z11-16:Al is 0.3:100; *Kehat and Dunkelblum, 1990*; *Zhang et al., 2012*), while sympatric *Heliothis peltigera* females emit a higher amount of the same component (ratio of Z9-14:Al to Z11-16:Al is 14.6:100; see *Hillier and Baker, 2016*). The secondary pheromone component, Z9-16:Al, resembles Z9-14:Al in that it may also include both enhancement and inhibition of attraction, dependent on the ratio. Concretely, Z9-16:Al is known to augment attraction when combined with the primary pheromone Z11-16:Al, but to reduce attraction when constituting a ratio to primary pheromone more than 11:100 (*Kehat and Dunkelblum, 1990*; *Kehat et al., 1980*). Notably, high ratios of Z9-16:Al may serve as an interspecific signal, due to it being the primary pheromone of sympatric *H. assulta* females (*Cork et al., 1992*). Altogether, in light of the complex behavioral functions of Z9-16:Al and Z9-14:Al, our data indicate that the anteroventral part of the LH is involved in inhibition of attraction when the moth encounters high ratios of the relevant components.

## A proposed interaction between the LH and the SLP/SIP

How can each of the two substances, Z9-14:Al and Z9-16:Al, serve opposite functions? During mate-searching behavior, the likelihood of encountering the primary pheromone is much larger than detecting the minor components, due to their proportion ratios in the natural female-released pheromone mixture. As such, the SLP and SIP, which receive information about the primary pheromone, is likely to be activated prior to the anteroventral LH. Notably, a prominent neural link between the LH and SLP in moths (*Namiki and Kanzaki, 2019*) appears to be a strong candidate for governing the 'correct' mate-searching behavior. A study in the fruit fly has demonstrated that the LH output projects primarily to SLP and secondly to the SIP, and about one-third of these neurons are GABAergic/glutamatergic (*Dolan et al., 2019*). Since both neurotransmitters are inhibitory (*Liu and Wilson, 2013*), this indicates that LH input may block the activation of SLP and SIP output neurons and their downstream circuits. The interaction between the LH and SLP/SIP could act as a Boolean logic gate implementing the AND-NOT function, only passing information downstream when aversive signaling is minimally present in the anteroventral LH.

We propose that this form of Boolean logic represents a flexible interaction between the LH and SLP/SIP. The LH output is mainly modulated by the input from the AL PNs, and the activity of these

neurons increases with increasing odor concentration (*Gupta and Stopfer, 2012*; *Lerner et al., 2020*; *Sachse and Galizia, 2003*). When a moth encounters the primary pheromone and a very low amount of the secondary pheromone (released by conspecific female), the LH input is weak and the inhibition from the LH to SLP/SIP minimal, therefore, the output from the SLP/SIP to downstream targets remains strong. Should a big amount of the secondary pheromone or the behavioral antagonist (released by a heterospecific female) recruit vigorous MGC-PN output onto the LH, the GABAergic/glutamatergic LH neurons will inhibit the activity in SLP/SIP and the downstream signaling. This might be perceived as a loss of contact with the primary pheromone, leading the moth to quickly engage in a casting flight, that is, crosswind flight with no net upwind movement (*Kuenen and Cardé, 1994*), and thereby promote an appropriate mate-searching strategy.

Given that the SLP/SIP is involved in processing pheromone information solely about attraction, while the LH represents ratio-dependent ambiguous signals, one question appearing is how the presence of a low dose of the minor components enhances attraction when added to the primary pheromone. Concerning Z9-16:Al, for example, our findings indicate that integration of the secondary and primary pheromone might occur at the level of MGC output PNs. Here, we first analyzed the physiological data of 11 uniglomerular medial-tract PNs with projections to SLP/SIP having excitatory responses to the primary pheromone and pheromone mixture. Their response amplitudes as well as response peaks were compared during stimulation with the primary pheromone alone and the binary mixture (ratio of secondary pheromone vs. primary pheromone ranged from 3:100 to 5:100). In both electrophysiological recording and calcium imaging experiments, the pheromone mixture showed a tendency of evoking a stronger response (*Figure 6C*). We also found at least three cumulus-PNs with projections in the SLP that had a tonic and delayed excitation to the secondary pheromone (*Figure 5* and *Figure 5—figure supplement 3G*) suggesting the existence of local computation.

## The pheromone-processing role of the VLP is complex

Unlike the neuropils discussed above, the VLP was innervated by all MGC-PNs labelled in this study. Specifically, they sent terminal projections into a compact sub-domain of the dorsoanterior VLP, which thereby gets intermingled signals about all female-relevant compounds. This suggests that the VLP is involved in combinatorial coding, possibly recognizing the optimal species-specific signal. In addition, combinatorial coding of pheromone signals seems to occur at an earlier level as well. The dendritic organization of the multiglomerular MGC-PNs prime these neurons for such signal processing. This was, perhaps, most clearly demonstrated by the broad response profiles of the mediolateral-tract PNs with evenly distributed dendrites across the MGC (*Figure 5* and *Figure 5—figure supplement 7*). These PNs responded to all female-produced components, but with distinct temporal patterns (*Figure 7*), and may provide excitatory or inhibitory output since about half of the axons forming this tract are GABAergic (*Berg et al., 2009*). Furthermore, we demonstrated that these mlALT PNs had overlapping terminals with the medial-tract MGC-PNs in this neuropil (*Figure 4A*), indicating the possibility of axo-axonic interaction. As the VLP is one known region with overlapping terminals of pheromone PNs across different tracts, this makes it a particularly interesting neuropil for future studies investigating parallel processing in the pheromone system.

# Materials and methods

## Insects

Male moths (2–3 days) of *H. armigera* were used in this study. Pupae were purchased from Keyun Bio-pesticides (Henan, China). After emergence, the moths were kept at 24°C and 70% humidity on a 14:10 hr light/dark cycle, with 10% sucrose solution available ad libitum. According to Norwegian law of animal welfare, there are no restrictions regarding experimental use of Lepidoptera.

## Calcium imaging

Totally, eight males (2–3 days) were used in calcium imaging experiments. Retrograde selective staining of MGC-PNs has been reported elsewhere (*Chu et al., 2020a*; *Ian et al., 2017*; *Sachse and Galizia, 2002*; *Sachse and Galizia, 2003*). A glass electrode tip coated with Fura-2 dextran (potassium salt, 10,000 MW, Molecular Probes) was inserted into the calyces to selectively label the

medial-tract PNs. The insects were then kept for 12 hr at 4°C before the experiment, to facilitate retrograde transportation.

In vivo calcium imaging recordings were obtained with an epifluorescent microscope (Olympus BX51WI) equipped with a 20x/1.00 water immersion objective (OlympusXLUMPlanFLN). Images were acquired by a 1344 × 1224 pixel CMOS camera (Hamamatsu ORCA-Flash4.0 V2 C11440-22CU). The preparation was excited with 340 and 380 nm monochromatic light, respectively (TILL Photonics Polychrome V), and data were acquired ratiometrically. A dichroic mirror (420 nm) and an emission filter (490–530 nm) were used to separate the excitation and emission light. Each recording consisted of 100 double frames at a sampling frequency of 10 Hz with 35 and 10 ms exposure times for the 340 and 380 nm lights, respectively. The duration of one recording trial was 10 s, including 4 s with spontaneous activity, 2 s odor stimulation, and a 4 s post-stimulus period. The odor stimulation was carried out by a stimulus controller (SYNTECH CS-55), via which humidified charcoal filtered air was delivered through a 150 mm glass Pasteur-pipette with a piece of filter paper containing the stimulus. Each odor stimulus was applied twice. The interval between trials was 60 s to avoid possible adaptation.

## Intracellular recording and staining

Preparation of the insect has been described in detail elsewhere (*Kc et al., 2020*; *Kymre et al., 2021*). Briefly, the moth was restrained inside a plastic tube with the head exposed and then immobilized with dental wax (Kerr Corporation, Romulus, MI). The brain was exposed by opening the head capsule and removing the muscle tissue. The exposed brain was continuously supplied with Ringer's solution (in mM): 150 NaCl, 3 CaCl$_2$, 3 KCl, 25 sucrose, and 10 N-tris (hydroxymethyl)-methyl-2-amino-ethanesulfonic acid, pH 6.9.

The procedure of intracellular recording/staining of neurons was performed as previously described (*Chu et al., 2020a*; *Ian et al., 2016*; *Zhao et al., 2014*). Sharp glass electrodes were made by pulling borosilicate glass capillaries (OD 1 mm, ID 0.5 mm, with hollow filament 0.13 mm; Hilgenberg GmbH, Germany) on a horizontal puller (P97; Sutter Instruments, Novarto, CA). The tip of the micro-pipette was filled with a fluorescent dye, that is, 4% biotinylated dextran-conjugated tetramethylrhodamine (3000 mw, micro-ruby, Molecular Probes; Invitrogen, Eugene, OR) in 0.2 M KAc. The glass capillary was back-filled with 0.2 M KAc. A chlorinated silver wire inserted into the muscle in the mouthpart served as the reference electrode. The recording electrode, having a resistance of 70–150 MΩ, was carefully inserted into the dorsolateral region of the AL via a micromanipulator (Leica). Neuronal spike activity was amplified (AxoClamp 1A, Axon Instruments, Union, CA) and monitored continuously by oscilloscope and loudspeaker. Spike2 6.02 (Cambridge Electronic Design, Cambridge, England) was used as acquisition software. During recording, the moth was ventilated constantly with a steady stream of fresh air. During odor stimulation, a pulse of air from the continuous airstream was diverted via a solenoid-activated valve (General Valve Corp., Fairfield, NJ) through a glass cartridge bearing the odorant on a piece of filter paper. Up to six odors were tested in each recording experiment, while the number of trials were dependent on the stability of the neuronal contact. The duration of the odor stimulus was 400 ms. After testing all odor stimuli, the neuron was iontophoretically stained by applying 2–3 nA pulses with 200 ms duration at 1 Hz for about 5–10 min. In order to allow neuronal transportation of the dye, the preparation was kept overnight at 4°C. The brain was then dissected from the head capsule and fixed in 4% paraformaldehyde for 1–2 hr at room temperature, before it was dehydrated in an ascending ethanol series (50%, 70%, 90%, 96%, 2 × 100%; 10 min each). Finally, the brain was cleared and mounted in methylsalicylate.

## Odor stimulation

During intracellular recordings, the following stimuli were tested: (i) the primary sex pheromone of *H. armigera*, Z11-16:Al, (ii) the secondary sex pheromone, Z9-16:Al, (iii) the binary mixture of Z11-16:Al and Z9-16:Al, (iv) the behavioral antagonist of *H. armigera*, Z9-14:Al, (v) the head space of a host plant (sunflower leaves), and (vi) hexane as a vehicle control. The three insect-produced components were obtained from Pherobank (Wijk bij Duurstede, Netherlands). Each component was diluted in hexane (99%, Sigma) at different concentrations before being applied onto a filter paper that was placed inside a 120 mm glass cartridge. Stimuli i–iv were used in two distinct stimulation protocols, that is, either with low or high concentrations of the respective stimuli. For the low concentration

protocol, 20 µl of the 0.5 ng/µl stimulus stock was applied onto the filter paper, making the final amount of 10 ng of the relevant stimulus. For the high concentration protocol, we applied 20 µl of the 5 ng/µl stimulus stock, resulting in a stimulus dosage of 100 ng per filter paper. In the low concentration protocol, the mixture of Z11-16:Al and Z9-16:Al were in a 100:5 proportion, while in the high concentration protocol it was in a 100:3 proportion, both resembling the natural blend emitted by conspecific females (*Hillier and Baker, 2016*; *Kehat et al., 1980*; *Piccardi et al., 1977*; *Wu et al., 1997*). Note that the ID of PNs stimulated with the low concentration protocol were numbered (e.g., Cu-mALT1), while the PNs stimulated with the high stimulation protocol were named with letters (e.g., Cu-mALTa). To avoid adaptation, PNs in the high concentration protocol did not receive repeated trials, as large dosages of pheromones are associated with prolonged adaptation of the relevant OSNs (*Dolzer et al., 2003*). The same odor stimuli as listed above were used during the calcium imaging experiments, but at higher amounts (required to evoke responses in this technique), that is, 10 µg at the filter paper. To avoid adaptation, the interstimulus interval was 1 min. An additional stimulus containing a 50:50 mixture of host plant (20 µl) and pheromone mix (20 µl 100:5 mixture) was added in the calcium imaging measurements.

## Confocal microscopy

Whole brains were imaged by using a confocal laser-scanning microscope (LSM 800 Zeiss, Jena, Germany) equipped with a Plan-Neofluar 20x/0.5 objective and/or a 10x/0.45 water objective (C-achroplan). Micro-ruby staining was excited with a HeNe laser at 553 nm and the fluorescent emission passed through a 560 nm long-pass filter. In addition to the fluorescent dyes, the auto-fluorescence of endogenous fluorophores in the neural tissue was imaged to visualize relevant neuropil structures in the brain containing the stained neurons. Since many fluorophore molecules are excited at 493 nm, these auto-fluorescent images were obtained by using an argon laser in combination with a 505–550 nm band pass filter. For both channels, serial optical sections with a resolution of 1024 × 1024 pixels were obtained at 2–8 µm intervals. Confocal images were edited in ZEN 2 (blue edition, Carl Zeiss Microscopy GmbH, Jana, Germany) and MATLAB R2018b.

## Reconstruction of individual neurons and nomenclature

To determine the targeted neuropil, each of the main MGC-PN types and its protocerebral target neuropils were reconstructed in AMIRA 5.3 (Visualization Science Group) by using the *SkeletonTree* plugin (*Evers et al., 2005*; *Schmitt et al., 2004*) and the segmentation tool, respectively. By this, the morphology of each neuron was digitally reproduced, including its neuronal filaments and thicknesses as well as its output regions. For visualization purpose, we plotted each traced skeleton onto the representative brain, based on the reconstructed data obtained by using the 'transform editor' function in AMIRA.

Regarding naming system, the 'delta region of the inferior lateral protocerebrum' (ΔILPC), a pyramid-shaped area formed by the male-specific medial- and mediolateral-tract PNs in the silk moth, *B. mori*, was previously described as a main MGC-output area (e.g., *Seki et al., 2005*). The current standard insect brain nomenclature, established by *Insect Brain Name Working Group et al., 2014*, does not include the ΔILPC, which seems to be part of several protocerebral neuropils (*Insect Brain Name Working Group et al., 2014*; *Lee et al., 2019*; *Lei et al., 2013*). The identification and naming of neuropil structures in the representative brain of *H. armigera* has already been adapted from the standard nomenclature established by *Insect Brain Name Working Group et al., 2014* and we utilize the same nomenclature here. The orientation of all brain structures is indicated relative to the body axis of the insect, as in *Homberg et al., 1988*.

## Calcium imaging data analysis

In this study, responses of medial-tract PNs innervating each MGC unit were analyzed. Recordings were acquired with Live Acquisition V2.3.0.18 (TILL Photonics GmbH, Kaufbeuren, Germany) and imported in KNIME Analytics Platform 2.12.2 (KNIME GmbH, Konstanz, Germany). Here, ImageBee neuro plugin (*Strauch et al., 2013*) was used to construct AL maps and glomerular time series. To determine an average baseline activity, the Fura signal representing the ratio between 340 and 380 nm excitation light ($F_{340}/F_{380}$) from 0.5 to 2.5 s (frames 5–25, within 4 s spontaneous activity) was selected and set to zero. Neuronal activity traces were thus represented as changes in fluorescent

level, specified as $\Delta F_{340}/F_{380}$. We defined a *Threshold* based upon the control (hexane) traces across eight individuals at a 5% significance level: $Threshold = Maximum\left(\overline{\Delta F_{340}/F_{380}} + 1.96\sigma_{\Delta F_{340}/F_{380}}\right)$. Responses were defined when the mean peak of $\Delta F_{340}/F_{380}$ trace (n = 8) for distinct stimuli was higher than *Threshold*. For displaying the response amplitude for each stimulus, the averaged $\Delta F_{340}/F_{380}$ within 2 s stimulation window was used.

## Spike data analysis

The electrophysiological data was spike-sorted and analyzed in Spike 2.8. Each odor application trial comprised a total period of 2.4 s, including a 1 s pre-stimulation window for baseline activity prior to the stimulus onset, 0.4 s stimulation period, and 1 s post-stimulation period. For describing the temporal neural activity, the Z-scored instantaneous firing rates (ZIFR) of every 10 ms for each trial were registered. To measure stimulus-specific responses of individual PNs, the odor-evoked response properties were analyzed in the mean ZIFR (MZIFR) across repetitive trials with the same stimulus. Significant responses were determined by the upper threshold ($T_U$) and lower threshold ($T_L$) calculated according to the mean MZIFR in the 1 s pre-stimulation window ($MZIFR_{PS}$) at a 5% significance level:

$$T_U \text{ of repetitive trials} = \overline{MZIFR_{PS}} + 1.96\sigma_{MZIFR_{PS}}$$

$$T_L \text{ of repetitive trials} = \overline{MZIFR_{PS}} - 1.96\sigma_{MZIFR_{PS}}$$

If the individual ZIFR in the stimulation window higher than the value of $T_U$ or lower than $T_L$, the stimulation was determined as evoking excitatory or inhibitory response, respectively.

To illustrate the response patterns, the stimulation window was divided into two sub-windows (SW), where the SW(i) included the first 100 ms of the stimulation period and SW(ii) the remaining 300 ms. If one of the mean ZIFRs during the stimulation sub-windows was higher than the value of 'Upper Threshold' or lower than 'Lower Threshold,' the change in activation was determined as an excitatory or inhibitory response, respectively.

For displaying the response amplitude for each stimulus, we first standardized the baseline activity by setting the MZIFR before stimulation onset to zero. The response amplitude was then quantified as the ΔMZIFR averaged within each of the two stimulation sub-windows. The ΔMZIFR in a 200 ms post-stimulation window was also computed. For each trial, the onset of an excitatory response was determined at the time point when the Z-scored instantaneous firing rate (binned every 1 ms) exceeded the corresponding response threshold (same formula as $T_U$). The mean responses of PNs within groups categorized by the dendritic or axonal innervations were also computed.

## Acknowledgements

We thank Stanley Heinze for his contribution to the representative brain atlas, Baiwei Ma for assistance with data collection, and Tom Knudsen for technical support. Funding This project was funded by the Norwegian Research Council, Project No. 287052, to BG Berg; the National Natural Science Foundation of China, Project No. 31861133019, to GR Wang; and the Program for Science and Technology Innovation Talents in Universities of Henan Province, Project No. 19HASTIT011.

## Additional information

### Funding

| Funder | Grant reference number | Author |
|---|---|---|
| Norges Forskningsråd | 287052 | Bente G Berg |
| National Natural Science Foundation of China | 31861133019 | GuiRong Wang |
| The Education Department of Henan Province | 19HASTIT011 | XinCheng Zhao |

The funders had no role in study design, data collection and interpretation, or the decision to submit the work for publication.

### Author contributions
Jonas Hansen Kymre, Formal analysis, Investigation, Visualization, Writing - original draft; XiaoLan Liu, Investigation, Visualization, Writing - review and editing; Elena Ian, Formal analysis, Visualization, Writing - review and editing; Christoffer Nerland Berge, Investigation, Writing - review and editing; GuiRong Wang, Resources, Funding acquisition; Bente Gunnveig Berg, Conceptualization, Resources, Supervision, Funding acquisition, Methodology, Writing - original draft, Project administration, Writing - review and editing; XinCheng Zhao, Conceptualization, Resources, Funding acquisition, Methodology, Writing - review and editing; Xi Chu, Conceptualization, Software, Formal analysis, Supervision, Investigation, Visualization, Methodology, Writing - original draft, Writing - review and editing

### Author ORCIDs
XinCheng Zhao ⓘD https://orcid.org/0000-0002-9471-2222
Xi Chu ⓘD https://orcid.org/0000-0002-0889-6345

### Ethics
Animal experimentation: According to Norwegian law of animal welfare, there are no restrictions regarding experimental use of Lepidoptera.

### Decision letter and Author response
Decision letter https://doi.org/10.7554/eLife.65683.sa1
Author response https://doi.org/10.7554/eLife.65683.sa2

## Additional files

### Supplementary files
• Transparent reporting form

### Data availability
All data generated or analysed during this study are included in the manuscript and supporting files.

The following dataset was generated:

| Author(s) | Year | Dataset title | Dataset URL | Database and Identifier |
|---|---|---|---|---|
| Kymre JH, Liu XL, Ian E, Berge CN, Wang GR, Berg BG, Zhao XC, Chu X | 2021 | MGC_PNs | https://insectbraindb.org/app/neuron/dataset/bca871ef-9f27-4128-ad5f-68ff15e5c7d7 | Insect Brain Database, bca871ef-9f27-4128-ad5f-68ff15e5c7d7 |

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
