## [Decision Letter]

**Acceptance summary:**

This study identifies and describes the functional properties of antennal lobe output neurons towards the response to pheromone odors in the moth brain. This work presents detailed functional evidence for the segregation of olfactory values in the moth brain, and will be of interest to neuroscientists investigating how sensory information is organized in the brain. Through a combination of technically challenging experiments, the study identifies the brain regions that differentially process attractive vs aversive olfactory pheromone signals. This foundational study provides compelling evidence for how the moth brain interprets complex pheromone olfactory odors.

**Decision letter after peer review:**

Thank you for submitting your article "Distinct protocerebral neuropils associated with attractive and aversive female-produced odorants in the male moth brain" for consideration by *eLife*. Your article has been reviewed by 3 peer reviewers, one of whom is a member of our Board of Reviewing Editors, and the evaluation has been overseen by Piali Sengupta as the Senior Editor. The following individual involved in review of your submission has agreed to reveal their identity: Monika Stengl (Reviewer #3).

Essential revisions:

1. Improve the writing and data presentation to better engage a broader audience and clarify the major findings. Jargon, abbreviations, background information, descriptions, etc need to be more clearly introduced and discussed. Also, data should be presented in the figures in a way that is easier to appreciate the major findings, especially for someone who is not an expert in moth PN physiology. For example (but not exclusively), the single neurons in Figure 2 could be incorporated into a standard brain and 3D activity patterns predicted based on the recordings. This might help in the presentation of the connection between PN anatomy and the activity dynamics of the 4 different odors.

2. A major concern is that the interpretations may not hold up if more PNs were tested. In other words, what is the completeness of the findings? If the authors were to perform another 30 recordings/fillings, would more types of PNs be found that would then change their interpretations? How often was the same PN labeled? We ask the authors to discuss how they have addressed this concern.

3. The study would benefit by the authors registering the PNs into a standard moth brain (as done in other insect species, such as honeybees and flies) to allow a categorization and matching of morphological properties. It appears in Figure 2 that these PN neurons might be ready for such analysis. In addition to helping clarify the connection between anatomy and function (point 1), this would also allow the different PNs to be compared and categorized based on morphological parameters and subsequently be assigned to specific neuron classes.

*Reviewer #1 (Recommendations for the authors):*

It would be helpful if the authors could present data regarding the completeness of their study. For example, how often was the same PN labeled? If the authors were to do another 30 recordings/fillings, how many more types of PNs might they find? How might this change their interpretations?

How similar are the protocerebrum innervation patterns of each PN type? It might be helpful to indicate innervation fields of the PNs as 3D heat-maps in the brain.

The mALT PNs appear to have high spontaneous firing rates (as shown in Figure 4- Sup 1). Some discussion of this, for example in comparison to fly PNs, is warranted. This noisy background makes the pheromone signals difficult to pickup.

The manuscript in its current form, might be more suitable for a specialized journal.

Figure 1. Please define the abbreviations used in A.

Please clarify in the text how fura-2 labeling from the calyx only labels the MGC PNs (Figure 1).

Figure 2. Please describe how the neurons were labeled in the figure legend.

*Reviewer #2 (Recommendations for the authors):*

In my opinion this study would largely improve and be less descriptive if the authors would assign the different PNs to morphological types based on neuronal reconstructions and registrations. These could then be linked to the functional properties obtained in the intracellular recordings. The reconstruction could furthermore be used to calculate and predict response patterns in the higher brain centers, such as the LH, VLP and SIP. In addition, I would somehow combine the calcium imaging data with the electrophysiological recordings to show that similar response properties could be observed for the PNs innervating the cumulus, dma and dmp compartment, respectively.

*Reviewer #3 (Recommendations for the authors):*

L39 Previous publications in Manduca sexta, *Apis mellifera*, *Drosophila melanogaster*, Papilio xuthus, and bumble bees indicated that there is no random neuron connectivity, but instead specific topographical distribution of visual and chemosensory inputs to the calyces of the mushroom body. Also, distinct arborization areas for pheromone- and general odor projection neuron inputs to calyces were described, as well as age-related/ task allocation-dependent antennal lobe to mushroom body projections. Furthermore, there is evidence for axonal subcompartments of Kenyon cells that are distinct in the processing of positive or negative valence. Please cite the respective publications and formulate more precisely what you intend to say about different connectivity schemes in the lateral horn versus the mushroom body.

L48-52 Please cite the work in Manduca sexta showing that precise ratios of pheromone components, as well as the duration and concentration of pheromone stimuli are very relevant for neuronal activity of antennal lobe output neurons.

L 66-75 Please explain very precisely at what concentration ratio the three different pheromone components tested occur in the pheromone blends of the two sympatric species *H. armigera* and *H. assulta*. Does a change in the ratio of primary pheromone component to third component determine repulsion, or, while keeping the correct ratio of the first, secondary and third pheromone components at a specific concentration threshold repulsion occurs? Please cite her published work for both species that is necessary to understand the effects of the three pheromone components on behavior. Name exact concentrations and ratios of one pheromone component to the other employed in your experiments. It is not clear whether >5% refers to the full blend as 100%….

L324…"primary pheromone component…

L326 A very simple scheme that shows the general arborization pattern of all three tracts would help, also summarizing the different names for the tracts published in the various species…which tract projects contralaterally, which tract projects first into the SIP, before it reaches the calyces? Are these tracts forming different delay lines…? Please describe the morphology of the respective tracts more precisely and compare with published tracts from other species within more detail as already done.

L389 Please be more precise and name respective concentrations and concentration ratios for each pheromone component.

L393.… on concentration or rather on a specific ratio with respect to the other pheromone components? Please distinguish throughout the manuscript…

L517 The duration of the odor stimulus was 400 ms.

L525 Since odor concentrations are very relevant for your experiments it is very important to tell exactly how much volume of which stock solution you employed and how you measured molar concentrations instead of only indicating doses applied to filter paper.

L528 did you mix components at the correct ratio of the natural pheromone blend? It did not become clear to me whether you changed two variables at the same time, the overall amplitude of the odor signal and the respective concentration ratios of the two compounds to each other in the pheromone mixtures.

L532-535 how can two different concentration ratios both resemble the natural blend? Please explain with reference to the two sympatric species…

Already in the introduction it should become clear which odor blend sympatric moth species use and what is known about the specificity of the respective olfactory receptor neurons and the information encoding of respective pheromone components in the antennal lobe.

L854 Please explain all abbreviations used in the Figure legends.

L856 Please replace "Pictorial material representing" with "Characteristic examples of". Delete "raw"

L858 Dashed white lines mark the AL, red lines the MGC.

L859 Calcium rises in response to…

L860 What are "standardized traces"? Are they normalized? Please explain.

Figure 1 and Suppl Figure 1: Are different sets of experiments used for Figure 1D, E and supplem. Figure 1? Or is the same set of experiments used for different data analysis?

Considering Figure legends:

Each Figure with its legend needs to be self-explanatory and should be understandable on its own without the need to read a reference or the main manuscript. Thus, in each Figure legend explain all abbreviations used. Also, name each significance test and the significance level for each star. Please include n=xxx number of experiments for each experiment not only in the main manuscript, but also in the Figure legends, It would be of considerable help for the non-specialist reader if each Figure legend contains its respective main conclusion, e.g. already in the first sentence.

---

## [Author Response]

Essential revisions:1. Improve the writing and data presentation to better engage a broader audience and clarify the major findings. Jargon, abbreviations, background information, descriptions, etc need to be more clearly introduced and discussed. Also, data should be presented in the figures in a way that is easier to appreciate the major findings, especially for someone who is not an expert in moth PN physiology. For example (but not exclusively), the single neurons in Figure 2 could be incorporated into a standard brain and 3D activity patterns predicted based on the recordings. This might help in the presentation of the connection between PN anatomy and the activity dynamics of the 4 different odors.

A comprehensive revision of all main parts of the manuscript was performed. We have, for example, included an introductive figure (Figure 1) providing essential background information. In the result section, we profoundly reorganized the data presentation by highlighting the major findings both in the text and figure material. As suggested by the editor, a new figure is made, figure 3 (substituting the original Figure 2), visualizing the main neuron types in separate panels as well as in joint plots (confocal data and 3D-models), and presenting descriptive/predictive frameworks reflecting the stimulus‑evoked neuronal activity within the relevant output regions of the PNs. The discussion is also reshaped, for instance, by including the issue of parallel olfactory processing in the current species as well as across different species. Altogether, we believe the revision has made the article more relevant to a broad audience. We hope our study dealing with one of the numerous insect species that inhabit our planet will be of interest.

2. A major concern is that the interpretations may not hold up if more PNs were tested. In other words, what is the completeness of the findings? If the authors were to perform another 30 recordings/fillings, would more types of PNs be found that would then change their interpretations? How often was the same PN labeled? We ask the authors to discuss how they have addressed this concern.

We thank the editor for highlighting the issue of data completeness. This issue has been well explained in the response to reviewer 1. A special paragraph in the discussion and new figure material, Figure 7D-G, were included in the revised manuscript.

3. The study would benefit by the authors registering the PNs into a standard moth brain (as done in other insect species, such as honeybees and flies) to allow a categorization and matching of morphological properties. It appears in Figure 2 that these PN neurons might be ready for such analysis. In addition to helping clarify the connection between anatomy and function (point 1), this would also allow the different PNs to be compared and categorized based on morphological parameters and subsequently be assigned to specific neuron classes.

Registering different PNs into a standard brain is clearly useful – especially if we want to compare the neurons’ projection patterns. However, errors due to local distortions often occur when registering neurons into a common brain frame. In fact, we have previous experience from affine/elastic registration of neurons into standard brains of both *H. assulta* and *H. virescens* male (Zhao et al., 2014). Another reliable source for such comparison is raw image data including identifiable neurons simultaneously stained in the same brain. In Figure 3C, we demonstrate overlapping terminal projections in the LH of two uniglomerular MGC-PNs originating from each of the two smaller MGC-units, the dma and dmp. And in Figure 4, we show the terminal projections of MGC-PNs confined to each of the three main tracts, demonstrating overlapping terminal arbors for medial- and mediolateral-tract neurons whereas the lateral-tract neuron projects to a separate area.

In line with the suggestion from the editor and one of the reviewers, we have added AMIRA reconstructions in the revised manuscript, including not only the skeleton of individual PNs but also 3D‑reconstructions of the neuropil regions innervated by each neuron. These data, confirming the neurons’ morphological properties, are presented in the figure supplement. In addition, for visualization purposes, we plotted each traced skeleton onto the representative brain, based on the reconstructed data obtained by using the ‘transform editor’ function in AMIRA (Figure 3). In addition, we have also submitted all morphological data (confocal stacks and 3D-AMIRA reconstructions) of the main MGC-PN types to the newly established Insect brain database (InsectbrainDB, 2021) for cross comparison with other insect species. For further details, see response to reviewer 2.

Reviewer #1 (Recommendations for the authors):It would be helpful if the authors could present data regarding the completeness of their study. For example, how often was the same PN labeled? If the authors were to do another 30 recordings/fillings, how many more types of PNs might they find? How might this change their interpretations?

We have added two new analyses, including a computational experiment that demonstrates the representativeness of the neurons presented here (see Figure 7D-G and discussion section Pheromone signaling along parallel antennal-lobe tracts).

How similar are the protocerebrum innervation patterns of each PN type? It might be helpful to indicate innervation fields of the PNs as 3D heat-maps in the brain.

The similarity of protocerebrual innervation is shown in Figure 3—figure supplement 1. The 3D maps are included in Figure 3.

The mALT PNs appear to have high spontaneous firing rates (as shown in Figure 4- Sup 1). Some discussion of this, for example in comparison to fly PNs, is warranted. This noisy background makes the pheromone signals difficult to pickup.

The issue was commented on in a distinct section of the discussion (see Pheromone signaling along parallel antennal-lobe tracts).

The manuscript in its current form, might be more suitable for a specialized journal.

We have revised the manuscript extensively, making the scientific content more accessible for a broad audience – still maintaining its relevance for more specialized scientists. We consider the issue covering neural pathways processing pheromone signals to be of general interest – especially when the data are obtained from a pest insect species which has a massive impact on human lives through its severe agricultural damage due to its high reproductive potential.

Figure 1. Please define the abbreviations used in A.

Defined (see Figure 2 legend).

Please clarify in the text how fura-2 labeling from the calyx only labels the MGC PNs (Figure 1).

This issue was clarified in Figure 2 legend.

Figure 2. Please describe how the neurons were labeled in the figure legend.

Described in detail in method (see Reconstruction of individual neurons and nomenclature) and briefly in the Figure 3 legend.

Reviewer #2 (Recommendations for the authors):In my opinion this study would largely improve and be less descriptive if the authors would assign the different PNs to morphological types based on neuronal reconstructions and registrations. These could then be linked to the functional properties obtained in the intracellular recordings. The reconstruction could furthermore be used to calculate and predict response patterns in the higher brain centers, such as the LH, VLP and SIP. In addition, I would somehow combine the calcium imaging data with the electrophysiological recordings to show that similar response properties could be observed for the PNs innervating the cumulus, dma and dmp compartment, respectively.

We have revised the manuscript extensively, for details see answer to editor, issue 3. The data from calcium imaging and electrophysiological recording were summarized in Figure 6 and the result section.

Reviewer #3 (Recommendations for the authors):L39 Previous publications in Manduca sexta, Apis mellifera, *Drosophila melanogaster*, Papilio xuthus, and bumble bees indicated that there is no random neuron connectivity, but instead specific topographical distribution of visual and chemosensory inputs to the calyces of the mushroom body. Also, distinct arborization areas for pheromone- and general odor projection neuron inputs to calyces were described, as well as age-related/ task allocation-dependent antennal lobe to mushroom body projections. Furthermore, there is evidence for axonal subcompartments of Kenyon cells that are distinct in the processing of positive or negative valence. Please cite the respective publications and formulate more precisely what you intend to say about different connectivity schemes in the lateral horn versus the mushroom body.

We agreed that the neuron connectivity in Ca is not “random”, but rather in a semi-random topographical distribution. As the reviewer pointed out, the signals in Ca are segregated according to the nature of the sensory input, e.g. pheromone and general odor PNs have their terminals located at the inner and outer part of calyces, respectively (Zhao et al., 2014), but connections between the projection neurons and the Kenyon cells are random (Caron et al., 2013). This background information were included in the Figure 1A legend.

L48-52 Please cite the work in Manduca sexta showing that precise ratios of pheromone components, as well as the duration and concentration of pheromone stimuli are very relevant for neuronal activity of antennal lobe output neurons.

The work in M. sexta was cited and a sentence describing other aspects of a pheromone plume was included.

L 66-75 Please explain very precisely at what concentration ratio the three different pheromone components tested occur in the pheromone blends of the two sympatric species H. armigera and H. assulta. Does a change in the ratio of primary pheromone component to third component determine repulsion, or, while keeping the correct ratio of the first, secondary and third pheromone components at a specific concentration threshold repulsion occurs? Please cite her published work for both species that is necessary to understand the effects of the three pheromone components on behavior. Name exact concentrations and ratios of one pheromone component to the other employed in your experiments. It is not clear whether >5% refers to the full blend as 100%….

Revised. The ratios of primary and secondary pheromone component to *H. armigera* and *H. assulta* were explained. The dual role of the third component was only reported in *H. armigera*, the relevant ratios were clearly described in the revised introduction.

L324…"primary pheromone component…

The paragraph was revised.

L326 A very simple scheme that shows the general arborization pattern of all three tracts would help, also summarizing the different names for the tracts published in the various species…which tract projects contralaterally, which tract projects first into the SIP, before it reaches the calyces? Are these tracts forming different delay lines…? Please describe the morphology of the respective tracts more precisely and compare with published tracts from other species within more detail as already done.

We have included a new figure in the introduction section, Figure 1, providing an overview of the olfactory pathway in male moths. Here, the schematic drawing (A) includes an overview of the uniglomerular medial-tract PNs confined to the plant-odor and pheromone sub-system, respectively, and their distinct paths from the periphery to the higher olfactory centers. In the schematic drawing (B), we provide an overview of the three main ALTs in the moth. A detailed description of the system is included in the relevant figure legend. In addition, we have included a section in the discussion that compares morphological and physiological properties of MGC- PNs confined to each of the three parallel tracts. Finally, a consideration implying the distinct roles of the parallel ALTs is added.

L389 Please be more precise and name respective concentrations and concentration ratios for each pheromone component.

Revised. The exact concentrations and ratios were mentioned through the entire manuscript.

L393.… on concentration or rather on a specific ratio with respect to the other pheromone components? Please distinguish throughout the manuscript…

Revised.

L517 The duration of the odor stimulus was 400 ms.

Revised.

L525 Since odor concentrations are very relevant for your experiments it is very important to tell exactly how much volume of which stock solution you employed and how you measured molar concentrations instead of only indicating doses applied to filter paper.

Added.

L528 did you mix components at the correct ratio of the natural pheromone blend? It did not become clear to me whether you changed two variables at the same time, the overall amplitude of the odor signal and the respective concentration ratios of the two compounds to each other in the pheromone mixtures.

Both ratios of primary to secondary component (100:5 and 100:3) are correct according to previous reports (for detail see explanation of the next issue). As the reviewer pointed out, the recording data was collected from two stimulation protocols: the ratio of 100:3 was used in 100 ng pheromone mixture and the ratio of 100:5 in 10 ng pheromone mixture. Before we decided to include the PNs’ responding data to the pheromone mixtures from both protocols, the spike data during the stimulation window (ZIFR trace comprising 40 bins) from 10 cumulus‑PNs was carefully analyzed, among them, 5 PNs sorted within the 100 ng protocol and 5 within the 10 ng protocol (Figure 5—figure supplement 2D). Repeated measures with one within factor (Bin) and one between factor (Protocol) were conducted on the spike data in 40 bins (binning size was 10 ms for 400ms) and between these two protocols. The analyses were then repeated in every 100 ms interval within the 400 ms stimulation window. The results showed there was no effect on the between-factor. Moreover, there was no interaction effect between the within-factor and the between-factor. The statistical data is listed in Author response table 1. We considered that too many detailed explorations possibly make the article less comprehensive for a broad audience, thus these statistical analyses were not included in the manuscript.

**Author response table 1. resptable1:** 

Factor	Protocol (Between two protocols)				
Interval in stimulation window	0-400 ms	0-100 ms	100-200 ms	200-300 ms	300-400 ms
bins	1-40	1-10	11-20	21-30	31-40
*df*	1	1	1	1	1
*error*	8	8	8	8	8
*F*	0.47	0.18	0.26	0.01	1.96
*p*	0.29	0.69	0.63	0.91	0.20
Factor	Protocol x Bin (Interaction effect of protocols and bins)				
Interval in stimulation window	0-400 ms	0-100 ms	100-200 ms	200-300 ms	300-400 ms
bins	1-40	1-10	11-20	21-30	31-40
*df*	39	9	9	9	9
*error*	312	72	72	72	72
*F*	0.56	0.42	0.92	1.00	0.79
*p*	0.99	0.92	0.52	0.45	0.63

L532-535 how can two different concentration ratios both resemble the natural blend? Please explain with reference to the two sympatric species…

The species-specific ratio is, strictly speaking, always defined as a range of ratios (see e.g. Kehat and Dunkelblum, 1990). These ratios could depend on a multitude of factors, including temperature, location, age, and host plant. Details on the sympatric species, armigera/assulta and armigera /peltigera, are mentioned in the revised manuscript .

Already in the introduction it should become clear which odor blend sympatric moth species use and what is known about the specificity of the respective olfactory receptor neurons and the information encoding of respective pheromone components in the antennal lobe.

After careful consideration, we decided to place this introductive information as in the original version. The reason is that this background knowledge is essential in order to understand the calcium imaging results, particularly for the non-specialized readers).

L854 Please explain all abbreviations used in the Figure legends.

Revised.

L856 Please replace "Pictorial material representing" with "Characteristic examples of". Delete "raw"

Revised.

L858 Dashed white lines mark the AL, red lines the MGC.

Revised.

L859 Calcium rises in response to…

Revised.

L860 What are "standardized traces"? Are they normalized? Please explain.

No, it was not normalized data. For the recorded fura signals measured during stimulation, we subtracted the average baseline to present the odor-induced changes of the calcium signal. This was clarified in the Figure 2C legend.

Figure 1 and Suppl Figure 1: Are different sets of experiments used for Figure 1D, E and supplem. Figure 1? Or is the same set of experiments used for different data analysis?

They are the same data set used for different analyses.

Considering Figure legends:Each Figure with its legend needs to be self-explanatory and should be understandable on its own without the need to read a reference or the main manuscript. Thus, in each Figure legend explain all abbreviations used. Also, name each significance test and the significance level for each star. Please include n=xxx number of experiments for each experiment not only in the main manuscript, but also in the Figure legends, It would be of considerable help for the non-specialist reader if each Figure legend contains its respective main conclusion, e.g. already in the first sentence.

Revised through all figure legends.